# Balancing the Scales: A Theoretical and Algorithmic Framework for Learning from Imbalanced Data

**Corinna Cortes** [1]  **Anqi Mao** [2]  **Mehryar Mohri** [1 2]  **Yutao Zhong** [1]

## Abstract

Class imbalance remains a major challenge in machine learning, especially in multi-class problems with long-tailed distributions. Existing methods, such as data resampling, cost-sensitive techniques, and logistic loss modifications, though popular and often effective, lack solid theoretical foundations. As an example, we demonstrate that cost-sensitive methods are not Bayes-consistent. This paper introduces a novel theoretical framework for analyzing generalization in imbalanced classification. We propose a new class-imbalanced margin loss function for both binary and multi-class settings, prove its strong $\mathcal{H}$-consistency, and derive corresponding learning guarantees based on empirical loss and a new notion of class-sensitive Rademacher complexity. Leveraging these theoretical results, we devise novel and general learning algorithms, IMMAX (*Imbalanced Margin Maximization*), which incorporate confidence margins and are applicable to various hypothesis sets. While our focus is theoretical, we also present extensive empirical results demonstrating the effectiveness of our algorithms compared to existing baselines.

## 1. Introduction

The class imbalance problem, defined by a significant disparity in the number of instances across classes within a dataset, is a common challenge in machine learning applications (Lewis & Gale, 1994; Fawcett & Provost, 1996; Kubat & Matwin, 1997; Kang et al., 2021; Menon et al., 2021; Liu et al., 2019; Cui et al., 2019). This issue is prevalent in many real-world binary classification scenarios, and arguably even more so in multi-class problems with numerous classes. In such cases, a few majority classes often dominate the dataset, leading to a "long-tailed" distribution. Classifiers trained on these imbalanced datasets often struggle on the minority classes, performing similarly to a naive baseline that simply predicts the majority class.

The problem has been widely studied in the literature (Cardie & Nowe, 1997; Kubat & Matwin, 1997; Chawla et al., 2002; He & Garcia, 2009; Wallace et al., 2011). While a comprehensive review is beyond our scope, we summarize key strategies into broad categories and refer readers to a recent survey by Zhang et al. (2023) for further details. The primary approaches include the following.

**Data modification methods.** Techniques such as oversampling the minority classes (Chawla et al., 2002), undersampling the majority classes (Wallace et al., 2011; Kubat & Matwin, 1997), or generating synthetic samples (e.g., SMOTE (Chawla et al., 2002; Qiao & Liu, 2008; Han et al., 2005)), aim to rebalance the dataset before training (Chawla et al., 2002; Estabrooks et al., 2004; Liu et al., 2008; Zhang & Pfister, 2021).

**Cost-sensitive techniques.** These assign different penalization costs to losses for different classes. They include cost-sensitive SVM (Iranmehr et al., 2019; Masnadi-Shirazi & Vasconcelos, 2010) and other cost-sensitive methods (Elkan, 2001; Zhou & Liu, 2005; Zhao et al., 2018; Zhang et al., 2018; 2019; Sun et al., 2007; Fan et al., 2017; Jamal et al., 2020). The weights are often determined by the relative number of samples in each class or a notion of effective sample size (Cui et al., 2019).

These two approaches are closely related and can be equivalent in the limit, with cost-sensitive methods offering a more efficient and principled implementation of data sampling. However, both approaches act by effectively modifying the underlying distribution and risk overfitting minority classes, discarding majority class information, and inherently biasing the training distribution. Very importantly, these techniques may lead to Bayes-inconsistency (proven in Section 6). So while effective in some cases, their performance depends on the problem, data distribution, predictors,

[1]Google Research, New York, NY; [2]Courant Institute of Mathematical Sciences, New York, NY. Correspondence to: Corinna Cortes <corinna@google.com>, Anqi Mao <aqmao@cims.nyu.edu>, Mehryar Mohri <mohri@google.com>, Yutao Zhong <yutaozhong@google.com>.

*Proceedings of the 42$^{nd}$ International Conference on Machine Learning*, Vancouver, Canada. PMLR 267, 2025. Copyright 2025 by the author(s).

and evaluation metrics (Van Hulse et al., 2007), and they often require extensive hyperparameter tuning. Hybrid approaches aim to combine these two techniques but inherit many of their limitations.

**Logistic loss modifications.** Several recent methods modify the logistic loss to address class imbalance. Some add hyperparameters to logits, effectively implementing cost-sensitive adjustments to the loss's exponential terms. Examples include the Balanced Softmax loss (Jiawei et al., 2020), Equalization loss (Tan et al., 2020), and LDAM loss (Cao et al., 2019). Other methods, such as logit adjustment (Menon et al., 2021; Khan et al., 2019), use hyperparameters for each pair of class labels, with Menon et al. (2021) showing calibration for their approach. Alternative multiplicative modifications were advocated by Ye et al. (2020), while the Vector-Scaling loss (Kini et al., 2021) integrates both additive and multiplicative adjustments. The authors analyze this approach for linear predictors, highlighting the specific advantages of multiplicative modifications. These multiplicative adjustments, however, are equivalent to normalizing scoring functions or feature vectors in linear cases, a widely used technique, regardless of class imbalance.

**Other methods.** Additional approaches for addressing imbalanced data (see (Zhang et al., 2023)) include post-hoc adjustments of decision thresholds (Fawcett & Provost, 1996; Collell et al., 2016) or class weights (Kang et al., 2020; Kim & Kim, 2020), and techniques like transfer learning, data augmentation, and distillation (Li et al., 2024b).

Despite the many significant advances, these techniques continue to face persistent challenges. Most existing solutions are heuristic-driven and lack a solid theoretical foundation, making their performance unpredictable across diverse contexts. Among prior work, the most closely related is that of (Cao et al., 2019), which provides an analysis of generalization guarantees for the *balanced loss*, which equalizes the impact of each class by weighting errors inversely to class frequency. Their analysis also applies only to binary classification under the separable case and does not address the target *misclassification loss*. Jiawei et al. (2020) adopt classical margin theory to derive generalization bounds for multi-class softmax regression, while Wang et al. (2023) establish fine-grained generalization bounds based on data-dependent contraction. Despite these advances, all of these works focus exclusively on the balanced loss. In contrast, our work establishes generalization guarantees with respect to the standard zero-one misclassification loss.

**Loss functions and fairness considerations.** This work focuses on the standard zero-one misclassification loss, which remains the primary objective in many machine learning applications. While the balanced loss is sometimes advocated for fairness, particularly when labels correlate with demographic attributes, such correlations are absent in many

tasks. Moreover, fairness involves broader considerations, and selecting the appropriate criterion requires complex trade-offs. Evaluation metrics like F1-score and AUC are also widely used in the context of imbalanced data. However, these metrics can obscure the model's performance on the standard zero-one misclassification tasks, especially in scenarios with extreme imbalances or when the minority class exhibits high variability.

**Our contributions.** This paper presents a comprehensive theoretical analysis of generalization for classification loss in the context of imbalanced classes.

In Section 3, we introduce a *class-imbalanced margin loss function* and provide a novel theoretical analysis for binary classification. We establish strong $\mathcal{H}$-consistency bounds and derive learning guarantees based on empirical class-imbalanced margin loss and class-sensitive Rademacher complexity. Section 4 details new learning algorithms, IM-MAX (*Imbalanced Margin Maximization*), inspired by our theoretical insights. These algorithms generalize margin-based methods by incorporating both positive and negative *confidence margins*. In the special case where the logistic loss is used, our algorithms can be viewed as a logistic loss modification method. However, they differ from previous approaches, including multiplicative logit modifications, as our parameters are applied multiplicatively to differences of logits, which naturally aligns with the concept of margins.

In Section 5, we extend our results to multi-class classification, introducing a generalized multi-class class-imbalanced margin loss, proving its $\mathcal{H}$-consistency, and deriving generalization bounds via confidence margin-weighted class-sensitive Rademacher complexity. We also present new IM-MAX algorithms for imbalanced multi-class problems based on these guarantees. In Section 6, we analyze two core methods for addressing imbalanced data. We prove that cost-sensitive methods lack Bayes-consistency and show that the analysis of Cao et al. (2019) in the separable binary case (for the balanced loss) leads to margin values conflicting with our theoretical results (for the misclassification loss). Finally, while the focus of our work is theoretical and algorithmic, Section 7 includes extensive empirical evaluations, comparing our methods against several baselines.

## 2. Preliminaries

**Binary classification.** Let $\mathcal{X}$ represent the input space, and $\mathcal{Y} = \{-1, +1\}$ the binary label space. Let $\mathcal{D}$ be a distribution over $\mathcal{X} \times \mathcal{Y}$, and $\mathcal{H}$ a hypothesis set of functions mapping from $\mathcal{X}$ to $\mathbb{R}$. Denote by $\mathcal{H}_{\text{all}}$ the set of all measurable functions, and by $\ell \colon \mathcal{H}_{\text{all}} \times \mathcal{X} \times \mathcal{Y} \to \mathbb{R}$ a loss function. The *generalization error* of a hypothesis $h \in \mathcal{H}$ and the *best-in-class generalization error* of $\mathcal{H}$ for a loss function $\ell$ are defined as follows: $\mathcal{R}_\ell(h) = \mathbb{E}_{(x,y)\sim\mathcal{D}}[\ell(h, x, y)]$, and

$\mathcal{R}_\ell^*(\mathcal{H}) = \inf_{h \in \mathcal{H}} \mathcal{R}_\ell(h)$. The target loss function in binary classification is the zero-one loss function defined for all $h \in \mathcal{H}$ and $(x, y) \in \mathcal{X} \times \mathcal{Y}$ by $\ell_{0-1}(h, x, y) \coloneqq 1_{\text{sign}(h(x)) \neq y}$, where $\text{sign}(\alpha) = 1_{\alpha \geq 0} - 1_{\alpha < 0}$. For a labeled example $(x, y) \in \mathcal{X} \times \mathcal{Y}$, the *margin* $\rho_h(x, y)$ of a predictor $h \in \mathcal{H}$ is defined by $\rho_h(x, y) = yh(x)$.

**Consistency.** A fundamental property of a surrogate loss $\ell_A$ for a target loss function $\ell_B$ is its *Bayes-consistency* (Zhang, 2004; Bartlett et al., 2006; Chen et al., 2004). Specifically, if a sequence of predictors $\{h_n\}_{n \in \mathbb{N}} \subset \mathcal{H}_{\text{all}}$ achieves the optimal $\ell_A$-loss asymptotically, then it also achieves the optimal $\ell_B$-loss in the limit: $\lim_{n \to +\infty} \mathcal{R}_{\ell_A}(h_n) = \mathcal{R}_{\ell_A}^*(\mathcal{H}_{\text{all}}) \Rightarrow \lim_{n \to +\infty} \mathcal{R}_{\ell_B}(h_n) = \mathcal{R}_{\ell_B}^*(\mathcal{H}_{\text{all}})$. While Bayes-consistency is a natural and desirable property, it is inherently asymptotic and applies only to the family of all measurable functions $\mathcal{H}_{\text{all}}$. A more applicable and informative notion is that of $\mathcal{H}$-*consistent bounds*, which account for the specific hypothesis class $\mathcal{H}$ and provide non-asymptotic guarantees (Awasthi et al., 2022a;b; 2021a;b; 2023a;b; Mao et al., 2023f;c;d;e;a;b; Zheng et al., 2023; Mao et al., 2024c;b;a;e;h;f;g; Mohri et al., 2024; Cortes et al., 2024; Mao et al., 2025a;b; Mao, 2025; Montreuil et al., 2024; 2025d;a;b;c; Zhong, 2025)). In the realizable setting, these bounds are of the form:

$$\forall h \in \mathcal{H}, \quad \mathcal{R}_{\ell_B}(h) - \mathcal{R}_{\ell_B}^*(\mathcal{H}) \leq \Gamma\big(\mathcal{R}_{\ell_A}(h) - \mathcal{R}_{\ell_A}^*(\mathcal{H})\big),$$

where $\Gamma$ is a non-increasing concave function with $\Gamma(0) = 0$. In the general non-realizable setting, each side of the bound is augmented with a *minimizability gap* $\mathcal{M}_\ell(\mathcal{H}) = \mathcal{R}_\ell^*(\mathcal{H}) - \mathbb{E}_x[\inf_{h \in \mathcal{H}} \mathbb{E}_y[\ell(h, x, y) \mid x]]$, which measures the difference between the best-in-class error and the expected best-in-class conditional error. The resulting bound is: $\mathcal{R}_{\ell_B}(h) - \mathcal{R}_{\ell_B}^*(\mathcal{H}) + \mathcal{M}_{\ell_B}(\mathcal{H}) \leq \Gamma\big(\mathcal{R}_{\ell_A}(h) - \mathcal{R}_{\ell_A}^*(\mathcal{H}) + \mathcal{M}_{\ell_A}(\mathcal{H})\big)$. $\mathcal{H}$-consistency bounds imply Bayes-consistency when $\mathcal{H} = \mathcal{H}_{\text{all}}$ (Mao et al., 2024i;d) and provide stronger and more applicable guarantees.

# 3. Theoretical Analysis of Imbalanced Binary Classification

Our theoretical analysis addresses imbalance by introducing distinct *confidence margins* for positive and negative points. This allows us to explicitly account for the effects of class imbalance. We begin by defining a general class-imbalanced margin loss function based on these confidence margins. Subsequently, we prove that, unlike previously studied cost-sensitive loss functions in the literature, this new loss function satisfies $\mathcal{H}$-consistency bounds. Furthermore, we establish general margin bounds for imbalanced binary classification in terms of the proposed class-imbalanced margin loss. While our use of margins bears some resemblance to the approach of Cao et al. (2019), their analysis is limited to *geometric margins* in the separable case, making

ours fundamentally distinct.

## 3.1. Imbalanced $(\rho_+, \rho_-)$-Margin Loss Function

We first extend the $\rho$-margin loss function (Mohri et al., 2018) to accommodate the imbalanced setting. To account for different confidence margins for instances with label $+$ and label $-$, we define the *class-imbalanced* $(\rho_+, \rho_-)$-*margin loss function* as follows:

**Definition 3.1** (Class-imbalanced margin loss function). Let $\Phi_\rho: u \mapsto \min\big(1, \max\big(0, 1 - \frac{u}{\rho}\big)\big)$ be the $\rho$-margin loss function. For any $\rho_+ > 0$ and $\rho_- > 0$, the *class-imbalanced* $(\rho_+, \rho_-)$-*margin loss* is the function $\mathsf{L}_{\rho_+, \rho_-}: \mathcal{H}_{\text{all}} \times \mathcal{X} \times \mathcal{Y} \to \mathbb{R}$, defined as follows:

$$\mathsf{L}_{\rho_+, \rho_-}(h, x, y) = \Phi_{\rho_+}(yh(x))1_{y=+1} + \Phi_{\rho_-}(yh(x))1_{y=-1}.$$

The main margin bounds in this section are expressed in terms of this loss function. The parameters $\rho_+$ and $\rho_-$, both greater than 0, represent the confidence margins imposed by a hypothesis $h$ for positive and negative instances, respectively. The following result provides an equivalent expression for the class-imbalanced margin loss function, see proof in Appendix D.1.

**Lemma 3.2.** *The class-imbalanced* $(\rho_+, \rho_-)$-*margin loss function can be equivalently expressed as follows:*

$$\mathsf{L}_{\rho_+, \rho_-}(h, x, y) = \Phi_{\rho_+}(yh(x))1_{h(x) \geq 0} + \Phi_{\rho_-}(yh(x))1_{h(x) < 0}.$$

## 3.2. $\mathcal{H}$-Consistency

The following result provides a strong consistency guarantee for the class-imbalanced margin loss introduced in relation to the zero-one loss. We say a hypothesis set is complete when the scoring values spanned by $\mathcal{H}$ for each instance cover $\mathbb{R}$: for all $x \in \mathcal{X}$, $\{h(x): h \in \mathcal{H}\} = \mathbb{R}$. Most hypothesis sets widely considered in practice are all complete.

**Theorem 3.3** ($\mathcal{H}$-consistency bound for class-imbalanced margin loss). *Let $\mathcal{H}$ be a complete hypothesis set. Then, for all $h \in \mathcal{H}$, $\rho_+ > 0$, and $\rho_- > 0$, the following bound holds:*

$$\begin{aligned} \mathcal{R}_{\ell_{0-1}}(h) &- \mathcal{R}_{\ell_{0-1}}^*(\mathcal{H}) + \mathcal{M}_{\ell_{0-1}}(\mathcal{H}) \\ &\leq \mathcal{R}_{\mathsf{L}_{\rho_+, \rho_-}}(h) - \mathcal{R}_{\mathsf{L}_{\rho_+, \rho_-}}^*(\mathcal{H}) + \mathcal{M}_{\mathsf{L}_{\rho_+, \rho_-}}(\mathcal{H}). \end{aligned}$$

The proof is presented in Appendix D.2. Note that our $\mathcal{H}$-consistency bounds in Theorem 3.3 can be extended directly to the uniformly bounded hypothesis sets considered in Theorem 4.1 below. In this case, the bounds would depend on the complexity of the hypothesis class, similar to the $\mathcal{H}$-consistency bounds presented in (Awasthi et al., 2022a). The next section presents generalization bounds based on the empirical class-imbalanced margin loss, along with the $(\rho_+, \rho_-)$-*class-sensitive Rademacher complexity* and its empirical counterpart defined below. Given a sample $S = $

$(x_1, \ldots, x_m)$, we define $I_+ = \{i \in \{1, \ldots, m\} \mid y_i = +1\}$ and $m_+ = |I_+|$ as the number of positive instances. Similarly, we define $I_- = \{i \in \{1, \ldots, m\} \mid y_i = -1\}$ and $m_- = |I_-|$ as the number of negative instances.

**Definition 3.4** (($\rho_+, \rho_-$)–class-sensitive Rademacher complexity)**.** Let $\mathcal{G}$ be a family of functions mapping from $\mathcal{Z}$ to $[a, b]$ and $S = (z_1, \ldots, z_m)$ a fixed sample of size $m$ with elements in $\mathcal{Z}$. Fix $\rho_+ > 0$ and $\rho_- > 0$. Then, the *empirical* ($\rho_+, \rho_-$)-*class-sensitive Rademacher complexity of* $\mathcal{G}$ with respect to the sample $S$ is defined as:

$$\widehat{\mathfrak{R}}_S^{\rho_+, \rho_-}(\mathcal{G}) = \frac{1}{m} \mathbb{E}_\sigma \left[ \sup_{g \in \mathcal{G}} \left\{ \sum_{i \in I_+} \frac{\sigma_i g(z_i)}{\rho_+} + \sum_{i \in I_-} \frac{\sigma_i g(z_i)}{\rho_-} \right\} \right],$$

where $\sigma = (\sigma_1, \ldots, \sigma_m)^\top$, with $\sigma_i$s independent uniform random variables taking values in $\{-1, +1\}$. For any integer $m \geq 1$, the ($\rho_+, \rho_-$)-*class-sensitive Rademacher complexity of* $\mathcal{G}$ is the expectation of the empirical ($\rho_+, \rho_-$)–class-sensitive Rademacher complexity over all samples of size $m$ drawn according to $\mathcal{D}$: $\mathfrak{R}_m^{\rho_+, \rho_-}(\mathcal{G}) = \mathbb{E}_{S \sim \mathcal{D}^m} \left[ \widehat{\mathfrak{R}}_S^{\rho_+, \rho_-}(\mathcal{G}) \right]$.

### 3.3. Margin-Based Guarantees

Next, we will prove a general margin-based generalization bound, which will serve as the foundation for deriving new algorithms for imbalanced binary classification.

Given a sample $S = (x_1, \ldots, x_m)$ and a hypothesis $h$, the *empirical class-imbalanced margin loss* is defined by $\widehat{\mathcal{R}}_S^{\rho_+, \rho_-}(h) = \frac{1}{m} \sum_{i=1}^m \mathsf{L}_{\rho_+, \rho_-}(h, x_i, y_i)$. Note that the zero-one loss function $\ell_{0-1}$ is upper-bounded by the class-imbalanced margin loss function $\mathsf{L}_{\rho_+, \rho_-}$: $\mathcal{R}_{\ell_{0-1}}(h) \leq \mathcal{R}_{\mathsf{L}_{\rho_+, \rho_-}}(h)$.

**Theorem 3.5** (Margin bound for imbalanced binary classification)**.** *Let $\mathcal{H}$ be a set of real-valued functions. Fix $\rho_+ > 0$ and $\rho_- > 0$, then, for any $\delta > 0$, with probability at least $1 - \delta$, each of the following holds for all $h \in \mathcal{H}$:*

$$\mathcal{R}_{\ell_{0-1}}(h) \leq \widehat{\mathcal{R}}_S^{\rho_+, \rho_-}(h) + 2\mathfrak{R}_m^{\rho_+, \rho_-}(\mathcal{H}) + \sqrt{\frac{\log \frac{1}{\delta}}{2m}}$$

$$\mathcal{R}_{\ell_{0-1}}(h) \leq \widehat{\mathcal{R}}_S^{\rho_+, \rho_-}(h) + 2\widehat{\mathfrak{R}}_S^{\rho_+, \rho_-}(\mathcal{H}) + 3\sqrt{\frac{\log \frac{2}{\delta}}{2m}}.$$

The proof is presented in Appendix D.3. The generalization bounds in Theorem 3.5 suggest a trade-off: increasing $\rho_+$ and $\rho_-$ reduces the complexity term (second term) but increases the empirical class-imbalanced margin loss $\widehat{\mathcal{R}}_S^{\rho_+, \rho_-}(h)$ (first term) by requiring higher confidence margins from the hypothesis $h$. Therefore, if the empirical class-imbalanced margin loss of $h$ remains small for relatively large values of $\rho_+$ and $\rho_-$, $h$ admits a particularly favorable guarantee on its generalization error.

For Theorem 3.5, the margin parameters $\rho_+$ and $\rho_-$ must be selected beforehand. But, the bounds of the theorem

can be generalized to hold uniformly for all $\rho_+ \in (0, 1]$ and $\rho_- \in (0, 1]$ at the cost of modest additional terms $\sqrt{\frac{\log \log_2 \frac{2}{\rho_+}}{m}}$ and $\sqrt{\frac{\log \log_2 \frac{2}{\rho_-}}{m}}$, as shown in Theorem D.1 in Appendix D.4.

## 4. Algorithm for Binary Classification

In this section, we derive algorithms for binary classification in imbalanced settings, building on the theoretical analysis from the previous section.

**Explicit guarantees.** Let $S \subseteq \{x \colon \|x\| \leq r\}$ denote a sample of size $m$. Define $r_+ = \sup_{i \in I_+} \|x_i\|$ and $r_- = \sup_{i \in I_-} \|x_i\|$. We assume that the empirical class-sensitive Rademacher complexity $\widehat{\mathfrak{R}}_S^{\rho_+, \rho_-}(\mathcal{H})$ can be bounded as:

$$\widehat{\mathfrak{R}}_S^{\rho_+, \rho_-}(\mathcal{H}) \leq \frac{\Lambda_{\mathcal{H}}}{m} \sqrt{\frac{m_+ r_+^2}{\rho_+^2} + \frac{m_- r_-^2}{\rho_-^2}} \leq \frac{\Lambda_{\mathcal{H}} r}{m} \sqrt{\frac{m_+}{\rho_+^2} + \frac{m_-}{\rho_-^2}},$$

where $\Lambda_{\mathcal{H}}$ depends on the complexity of the hypothesis set $\mathcal{H}$. This bound holds for many commonly used hypothesis sets. As an example, for a family of neural networks, $\Lambda_{\mathcal{H}}$ can be expressed as a Frobenius norm (Cortes et al., 2017; Neyshabur et al., 2015) or spectral norm complexity with respect to reference weight matrices (Bartlett et al., 2017). More generally, for the analysis that follows, we will assume that $\mathcal{H}$ can be defined by $\mathcal{H} = \{h \in \overline{\mathcal{H}} \colon \|h\| \leq \Lambda_{\mathcal{H}}\}$, for some appropriate norm $\|\cdot\|$ on some space $\overline{\mathcal{H}}$. For the class of linear hypotheses with bounded weight vector, $\mathcal{H} = \{x \mapsto w \cdot x \colon \|w\| \leq \Lambda\}$, we provide the following explicit guarantee. The proof is presented in Appendix D.6.

**Theorem 4.1.** *Let $S \subseteq \{x \colon \|x\| \leq r\}$ be a sample of size $m$ and let $\mathcal{H} = \{x \mapsto w \cdot x \colon \|w\| \leq \Lambda\}$. Let $r_+ = \sup_{i \in I_+} \|x_i\|$ and $r_- = \sup_{i \in I_-} \|x_i\|$. Then, the following bound holds for all $h \in \mathcal{H}$:*

$$\widehat{\mathfrak{R}}_S^{\rho_+, \rho_-}(\mathcal{H}) \leq \frac{\Lambda}{m} \sqrt{\frac{m_+ r_+^2}{\rho_+^2} + \frac{m_- r_-^2}{\rho_-^2}} \leq \frac{\Lambda r}{m} \sqrt{\frac{m_+}{\rho_+^2} + \frac{m_-}{\rho_-^2}}.$$

Combining the upper bound of Theorem 4.1 and Theorem 3.5 gives directly the following general margin bound:

$$\mathcal{R}_{\ell_{0-1}}(h) \leq \widehat{\mathcal{R}}_S^{\rho_+, \rho_-}(h) + \frac{2\Lambda_{\mathcal{H}}}{m} \sqrt{\frac{m_+ r_+^2}{\rho_+^2} + \frac{m_- r_-^2}{\rho_-^2}} + 3\sqrt{\frac{\log \frac{2}{\delta}}{2m}}.$$

As with Theorem 3.5, this bound can be generalized to hold uniformly for all $\rho_+ \in (0, 1]$ and $\rho_- \in (0, 1]$ at the cost of additional terms $\sqrt{\frac{\log \log_2 \frac{2}{\rho_+}}{m}}$ and $\sqrt{\frac{\log \log_2 \frac{2}{\rho_-}}{m}}$ by combining the bound on the class-sensitive Rademacher complexity and Theorem D.1. The bound suggests that a small generalization error can be achieved when the second term $\frac{\Lambda_{\mathcal{H}}}{m} \sqrt{\frac{m_+ r_+^2}{\rho_+^2} + \frac{m_- r_-^2}{\rho_-^2}}$ or $\frac{\Lambda_{\mathcal{H}} r}{m} \sqrt{\frac{m_+}{\rho_+^2} + \frac{m_-}{\rho_-^2}}$ is small while the

empirical class-imbalanced margin loss (first term) remains low.

Now, consider a margin-based loss function $(h, x, y) \mapsto \Psi(yh(x))$ defined using a non-increasing convex function $\Psi$ such that $\Phi_\rho(u) \leq \Psi\left(\frac{u}{\rho}\right)$ for all $u \in \mathbb{R}$. Examples of such $\Psi$ include: the hinge loss, $\Psi(u) = \max(0, 1 - u)$, the logistic loss, $\Psi(u) = \log_2(1 + e^{-u})$, and the exponential loss, $\Psi(u) = e^{-u}$.

Then, choosing $\Lambda_{\mathcal{H}} = 1$, with probability at least $1 - \delta$, the following holds for all $h \in \left\{h \in \overline{\mathcal{H}} \colon \|h\| \leq 1\right\}$, $\rho_+ \in (0, r_+]$ and $\rho_- \in (0, r_-]$:

$$\mathcal{R}_{\ell_{0-1}}(h) \leq \frac{1}{m}\left[\sum_{i \in I_+} \Psi\left(\frac{y_i h(x_i)}{\rho_+}\right) + \sum_{i \in I_-} \Psi\left(\frac{y_i h(x_i)}{\rho_-}\right)\right]$$
$$+ \frac{4r}{m}\sqrt{\frac{m_+}{\rho_+^2} + \frac{m_-}{\rho_-^2}} + O\left(\frac{1}{\sqrt{m}}\right),$$

where the last term includes the $\log$-$\log$ terms and the $\delta$-confidence term.

Since for any $\rho > 0$, $h/\rho$ admits the same generalization error as $h$, with probability at least $1 - \delta$, the following holds for all $h \in \left\{h \in \overline{\mathcal{H}} \colon \|h\| \leq \frac{1}{\rho_+ + \rho_-}\right\}$, $\rho_+$ and $\rho_-$:

$$\mathcal{R}_{\ell_{0-1}}(h) \leq \frac{1}{m}\left[\sum_{i \in I_+} \Psi\left(y_i h(x_i)\frac{\rho_+ + \rho_-}{\rho_+}\right)\right.$$
$$\left. + \sum_{i \in I_-} \Psi\left(y_i h(x_i)\frac{\rho_+ + \rho_-}{\rho_-}\right)\right] + \frac{4r}{m}\sqrt{\frac{m_+}{\rho_+^2} + \frac{m_-}{\rho_-^2}} + O\left(\frac{1}{\sqrt{m}}\right).$$

**Algorithm.** Now, since only the first term of the right-hand side depends on $h$, the bound suggests selecting $h$, with $\|h\|^2 \leq \left(\frac{1}{\rho_+ + \rho_-}\right)^2$ as a solution of:

$$\min_{h \in \overline{\mathcal{H}}} \frac{1}{m}\left[\sum_{i \in I_+} \Psi\left(y_i h(x_i)\frac{\rho_+ + \rho_-}{\rho_+}\right) + \sum_{i \in I_-} \Psi\left(y_i h(x_i)\frac{\rho_+ + \rho_-}{\rho_-}\right)\right].$$

Introducing a Lagrange multiplier $\lambda \geq 0$ and a free variable $\alpha = \frac{\rho_+}{\rho_+ + \rho_-} > 0$, the optimization problem can be written as

$$\min_{h \in \overline{\mathcal{H}}} \lambda\|h\|^2 + \frac{1}{m}\left[\sum_{i \in I_+} \Psi\left(\frac{h(x_i)}{\alpha}\right) + \sum_{i \in I_-} \Psi\left(\frac{-h(x_i)}{1 - \alpha}\right)\right], \quad (1)$$

where $\lambda$ and $\alpha$ can be selected via cross-validation.

This formulation provides a general algorithm for binary classification in imbalanced settings, called IMMAX (*Imbalanced Margin Maximization*), supported by strong theoretical guarantees derived in the previous section. This provides a solution for optimizing the decision boundaries in imbalanced settings based on confidence margins. In the specific case of linear hypotheses (Appendix D.5), choosing $\Psi$ as the Hinge loss yields a strict generalization of the SVM

algorithm which can be used with positive definite kernels, or a strict generalization of the logistic regression algorithm when $\Psi$ defines the logistic loss.

Beyond linear models, this algorithm readily extends to neural networks with various regularization terms and other complex hypothesis sets. This makes it a general solution for tackling imbalanced binary classification problems.

**Separable case**. When the training sample is separable, we can denote by $\rho_{\text{geom}}$ the geometric margin, that is the smallest distance of a training sample point to the decision boundary measured in the Euclidean distance or another metric appropriate for the feature space. As an example, for linear hypotheses, $\rho_{\text{geom}}$ corresponds to the familiar Euclidean distance to the separating hyperplane.

The confidence margin parameters $\rho_+$ and $\rho_-$ can then be chosen so that $\rho_+ + \rho_- = 2\rho_{\text{geom}}$, ensuring that the empirical class-imbalanced margin loss term is zero. Minimizing the right-hand side of the bound then yields the following expressions for $\rho_+$ and $\rho_-$:

$$\rho_+ = \frac{2m_+^{\frac{1}{3}} r_+^{\frac{2}{3}}}{m_+^{\frac{1}{3}} r_+^{\frac{2}{3}} + m_-^{\frac{1}{3}} r_-^{\frac{2}{3}}}\rho_{\text{geom}} \quad \rho_- = \frac{2m_-^{\frac{1}{3}} r_-^{\frac{2}{3}}}{m_+^{\frac{1}{3}} r_+^{\frac{2}{3}} + m_-^{\frac{1}{3}} r_-^{\frac{2}{3}}}\rho_{\text{geom}}.$$

For $r_+ = r_-$, these expressions simplify to:

$$\rho_+ = \frac{2m_+^{\frac{1}{3}}}{m_+^{\frac{1}{3}} + m_-^{\frac{1}{3}}}\rho_{\text{geom}} \qquad \rho_- = \frac{2m_-^{\frac{1}{3}}}{m_+^{\frac{1}{3}} + m_-^{\frac{1}{3}}}\rho_{\text{geom}}. \quad (2)$$

Note that the optimal positive margin $\rho_+$ is larger than the negative one $\rho_-$ when there are more positive samples than negative ones ($m_+ > m_-$). Thus, in the linear case, this suggests selecting a hyperplane with a large positive margin in that case, see Figure 1 for an illustration.

Finally, note that, while $\alpha = \frac{\rho_+}{\rho_+ + \rho_-} > 0$ in the optimization problem (1) can be freely searched over a range of values in our general (non-separable case) algorithm, it can be beneficial to focus the search around the optimal values identified in the separable case.

## 5. Extension to Multi-Class Classification

In this section, we extend our results to multi-class classification, with full details provided in Appendix E. Below, we present a concise overview.

We will adopt the same notation and definitions as previously described, with some slight adjustments. In particular, we denote the multi-class label space by $\mathcal{Y} = [c] \coloneqq \{1, \ldots, c\}$ and a hypothesis set of functions mapping from $\mathcal{X} \times \mathcal{Y}$ to $\mathbb{R}$ by $\mathcal{H}$. For a hypothesis $h \in \mathcal{H}$, the label $\mathsf{h}(x)$ assigned to $x \in \mathcal{X}$ is the one with the largest score, defined as $\mathsf{h}(x) = \operatorname{argmax}_{y \in \mathcal{Y}} h(x, y)$, using the highest index for tie-breaking. For a labeled example $(x, y) \in \mathcal{X} \times \mathcal{Y}$,

the *margin* $\rho_h(x, y)$ of a hypothesis $h \in \mathcal{H}$ is given by $\rho_h(x, y) = h(x, y) - \max_{y' \neq y} h(x, y')$, which is the difference between the score assigned to $(x, y)$ and that of the next-highest scoring label. We define the multi-class zero-one loss function as $\ell_{0-1}^{\text{multi}} \coloneqq 1_{h(x) \neq y}$. This is the target loss of interest in multi-class classification.

We define the *multi-class class-imbalanced margin loss function* as follows:

**Definition 5.1** (Multi-class class-imbalanced margin loss)**.** For any $\boldsymbol{\rho} = [\rho_k]_{k \in [c]}$, the *multi-class class-imbalanced $\boldsymbol{\rho}$-margin loss* is the function $\mathsf{L}_{\boldsymbol{\rho}} \colon \mathcal{H}_{\text{all}} \times \mathcal{X} \times \mathcal{Y} \to \mathbb{R}$, defined by:

$$\mathsf{L}_{\boldsymbol{\rho}}(h, x, y) = \sum_{k=1}^{c} \Phi_{\rho_k}(\rho_h(x, y)) 1_{y=k}. \quad (3)$$

The main margin bounds in this section are expressed in terms of this loss function. The parameters $\rho_k > 0$, for $k \in [c]$, represent the confidence margins imposed by a hypothesis $h$ for instances labeled $k$. As in the binary case, we establish an equivalent expression for this class-imbalanced margin loss function.

**Lemma 5.2.** *The multi-class class-imbalanced $\rho$-margin loss can be equivalently expressed as follows:*

$$\mathsf{L}_{\boldsymbol{\rho}}(h, x, y) = \sum_{k=1}^{c} \Phi_{\rho_k}(\rho_h(x, y)) 1_{h(x)=k}.$$

The proof is included in Appendix F.1. We also prove that our multi-class class-imbalanced $\boldsymbol{\rho}$-margin loss is $\mathcal{H}$-consistent for any *complete* hypothesis set $\mathcal{H}$. We say a hypothesis set is complete when the scoring values spanned by $\mathcal{H}$ for each instance cover $\mathbb{R}$: for all $(x, y) \in \mathcal{X} \times \mathcal{Y}$, $\{h(x, y) \colon h \in \mathcal{H}\} = \mathbb{R}$. This covers all commonly used function classes in practice, such as linear classifiers and neural network architectures.

**Theorem 5.3** ($\mathcal{H}$-Consistency bound for multi-class class-imbalanced margin loss)**.** *Let $\mathcal{H}$ be a complete hypothesis set. Then, for all $h \in \mathcal{H}$ and $\boldsymbol{\rho} = [\rho_k]_{k \in [c]} > \mathbf{0}$, the following bound holds:*

$$\mathcal{R}_{\ell_{0-1}^{\text{multi}}}(h) - \mathcal{R}_{\ell_{0-1}^{\text{multi}}}^*(\mathcal{H}) + \mathcal{M}_{\ell_{0-1}^{\text{multi}}}(\mathcal{H})$$
$$\leq \mathcal{R}_{\mathsf{L}_{\boldsymbol{\rho}}}(h) - \mathcal{R}_{\mathsf{L}_{\boldsymbol{\rho}}}^*(\mathcal{H}) + \mathcal{M}_{\mathsf{L}_{\boldsymbol{\rho}}}(\mathcal{H}).$$

The proof is included in Appendix F.2. Our generalization bounds are expressed in terms of the following notions of $\boldsymbol{\rho}$-class-sensitive Rademacher complexity.

**Definition 5.4** ($\boldsymbol{\rho}$-class-sensitive Rademacher complexity)**.** Let $\mathcal{H}$ be a family of functions mapping from $\mathcal{X} \times \mathcal{Y}$ to $\mathbb{R}$ and $S = ((x_1, y_1) \dots, (x_m, y_m))$ a fixed sample of size $m$ with elements in $\mathcal{X} \times \mathcal{Y}$. Fix $\boldsymbol{\rho} = [\rho_k]_{k \in [c]} > \mathbf{0}$. Then, the *empirical $\boldsymbol{\rho}$-class-sensitive Rademacher complexity of $\mathcal{H}$*

with respect to the sample $S$ is defined as:

$$\widehat{\mathfrak{R}}_S^{\boldsymbol{\rho}}(\mathcal{H}) = \frac{1}{m} \mathbb{E}_{\epsilon} \left[ \sup_{h \in \mathcal{H}} \left\{ \sum_{k=1}^{c} \sum_{i \in I_k} \sum_{y \in \mathcal{Y}} \epsilon_{iy} \frac{h(x_i, y)}{\rho_k} \right\} \right], \quad (4)$$

where $\epsilon = (\epsilon_{iy})_{i, y}$ with $\epsilon_{iy}$s being independent variables uniformly distributed over $\{-1, +1\}$. For any integer $m \geq 1$, the *$\boldsymbol{\rho}$-class-sensitive Rademacher complexity of $\mathcal{H}$* is the expectation of the empirical $\boldsymbol{\rho}$-class-sensitive Rademacher complexity over all samples of size $m$ drawn according to $\mathcal{D}$: $\mathfrak{R}_m^{\boldsymbol{\rho}}(\mathcal{H}) = \mathbb{E}_{S \sim \mathcal{D}^m}[\widehat{\mathfrak{R}}_S^{\boldsymbol{\rho}}(\mathcal{H})]$.

**Margin bound.** We establish a general multi-class margin-based generalization bound in terms of the empirical multi-class class-imbalanced $\boldsymbol{\rho}$-margin loss and the empirical $\boldsymbol{\rho}$-class-sensitive Rademacher complexity (Theorem E.3). The bound takes the following form:

$$\mathcal{R}_{\ell_{0-1}^{\text{multi}}}(h) \leq \widehat{\mathfrak{R}}_S^{\boldsymbol{\rho}}(h) + 4\sqrt{2c} \, \mathfrak{R}_m^{\boldsymbol{\rho}}(\mathcal{H}) + O(1/\sqrt{m}).$$

This serves as the foundation for deriving new algorithms for imbalanced multi-class classification.

**Explicit guarantees.** Let $\Phi$ be a feature mapping from $\mathcal{X} \times \mathcal{Y}$ to $\mathbb{R}^d$. Let $S \subseteq \{(x, y) \colon \|\Phi(x, y)\| \leq r\}$ denote a sample of size $m$, for some appropriate norm $\|\cdot\|$ on $\mathbb{R}^d$. Define $r_k = \sup_{i \in I_k, y \in \mathcal{Y}} \|\Phi(x_i, y)\|$, for any $k \in [c]$. As in the binary case, we assume that the empirical class-sensitive Rademacher complexity $\widehat{\mathfrak{R}}_S^{\boldsymbol{\rho}}(\mathcal{H})$ can be bounded as:

$$\widehat{\mathfrak{R}}_S^{\boldsymbol{\rho}}(\mathcal{H}) \leq \frac{\Lambda_{\mathcal{H}} \sqrt{c}}{m} \sqrt{\sum_{k=1}^{c} \frac{m_k r_k^2}{\rho_k^2}} \leq \frac{\Lambda_{\mathcal{H}} r \sqrt{c}}{m} \sqrt{\sum_{k=1}^{c} \frac{m_k}{\rho_k^2}},$$

where $\Lambda_{\mathcal{H}}$ depends on the complexity of the hypothesis set $\mathcal{H}$. This bound holds for many commonly used hypothesis sets. For a family of neural networks, $\Lambda_{\mathcal{H}}$ can be expressed as a Frobenius norm (Cortes et al., 2017; Neyshabur et al., 2015) or spectral norm complexity with respect to reference weight matrices (Bartlett et al., 2017). Additionally, Theorems F.3 and F.4 in Appendix F.6 address kernel-based hypotheses. More generally, for the analysis that follows, we will assume that $\mathcal{H}$ can be defined by $\mathcal{H} = \{h \in \overline{\mathcal{H}} \colon \|h\| \leq \Lambda_{\mathcal{H}}\}$, for some appropriate norm $\|\cdot\|$ on some space $\overline{\mathcal{H}}$. Combining such an upper bound and Theorem E.3 or Theorem F.2, gives directly the following general margin bound:

$$\mathcal{R}_{\ell_{0-1}^{\text{multi}}}(h) \leq \widehat{\mathfrak{R}}_S^{\boldsymbol{\rho}}(h) + \frac{4\sqrt{2} \Lambda_{\mathcal{H}} rc}{m} \sqrt{\sum_{k=1}^{c} \frac{m_k}{\rho_k^2}} + O\left(\frac{1}{\sqrt{m}}\right),$$

where the last term includes the log-log terms and the $\delta$-confidence term. Let $\Psi$ be a non-increasing convex function such that $\Phi_\rho(u) \leq \Psi\left(\frac{u}{\rho}\right)$ for all $u \in \mathbb{R}$. Then, since $\Phi_\rho$ is non-increasing, for any $(x, k)$, we have: $\Phi_\rho(\rho_h(x, k)) = \max_{j \neq k} \Phi_\rho(h(x, k) - h(x, j))$.

**Algorithm.** This suggests a regularization-based algorithm of the following form:

$$\min_{h\in\overline{\mathcal{H}}}\lambda\|h\|^2 + \frac{1}{m}\left[\sum_{k=1}^{c}\sum_{i\in I_k}\max_{j\neq k}\Psi\left(\frac{h(x,k)-h(x,j)}{\rho_k}\right)\right], \quad (5)$$

where $\lambda$ and $\rho_k$s are chosen via cross-validation. In particular, choosing $\Psi$ to be the logistic loss and upper-bounding the maximum by a sum yields the following form for our IMMAX (*Imbalanced Margin Maximization*) algorithm:

$$\min_{h\in\overline{\mathcal{H}}}\lambda\|h\|^2 + \frac{1}{m}\sum_{k=1}^{c}\sum_{i\in I_k}\log\left[\sum_{j=1}^{c}\exp\left(\frac{h(x_i,j)-h(x_i,k)}{\rho_k}\right)\right], \quad (6)$$

where $\lambda$ and $\rho_k$s are chosen via cross-validation. Let $\rho = \sum_{k=1}^{c}\rho_k$ and $\overline{r} = \left[\sum_{k=1}^{c}m_k^{\frac{1}{3}}r_{k,2}^{\frac{2}{3}}\right]^{\frac{3}{2}}$, where as defined previously, $r_k = \sup_{i\in I_k, y\in\mathcal{Y}}\|\Phi(x_i,y)\|$, for any $k\in[c]$. Using Lemma F.1 (Appendix F.4), the term under the square root in the second term of the generalization bound can be reformulated in terms of the Rényi divergence of order 3 as: $\sum_{k=1}^{c}\frac{m_k r_{k,2}^2}{\rho_k^2} = \frac{\overline{r}^2}{\rho^2}e^{2\mathsf{D}_3\left(\mathsf{r}\,\|\,\frac{\rho}{\rho}\right)}$, where $\mathsf{r} = \left[\frac{m_k^{\frac{1}{3}}r_{k,2}^{\frac{2}{3}}}{\overline{r}^{\frac{2}{3}}}\right]_k$. Thus, while $\rho_k$s can be freely searched over a range of values in our general algorithm (6), for experimental efficiency it may be beneficial to focus the search for the vector $[\rho_k/\rho]_k$ near r. When the number of classes $c$ is very large, the search space can also be significantly reduced by assigning identical $\rho_k$ values to underrepresented classes while reserving distinct $\rho_k$ values for the most frequently occurring classes.

## 6. Formal Analysis of Some Core Methods

This section analyzes two popular methods presented in the literature for tackling imbalanced data.

**Resampling or cost-sensitive loss minimization.** A common approach for handling imbalanced data in practice is to assign distinct costs to positive and negative samples. This technique, implemented either explicitly or through resampling, is widely used in empirical studies (Chawla et al., 2002; He & Garcia, 2009; He & Ma, 2013; Huang et al., 2016; Buda et al., 2018; Cui et al., 2019). The associated target loss $\mathsf{L}_{c_+,c_-}(h,x,y)$ can be expressed as follows, for any $c_+ > 0$, $c_- > 0$ and $(h,x,y)\in\mathcal{H}_{\text{all}}\times\mathcal{X}\times\mathcal{Y}$:

$$c_+\ell_{0-1}(h,x,y)1_{y=+1} + c_-\ell_{0-1}(h,x,y)1_{y=-1}.$$

The following negative result, see also Appendix C, shows that this loss function does not benefit from a consistency, a motivating factor for our study of the class-imbalanced margin loss, Section 3, with strong consistency guarantees.

**Theorem 6.1** (Negative results for resampling and cost-sensitive methods)**.** *If $c_+ \neq c_-$, then $\mathsf{L}_{c_+,c_-}$ is not Bayes-consistent with respect to $\ell_{0-1}$.*

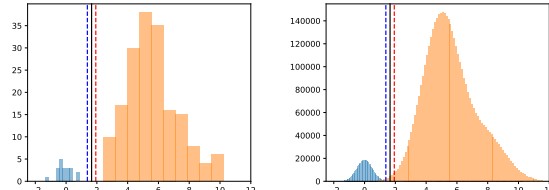

*Figure 1.* Solutions in the separable case. Left: Empirical data with negative (blue) and positive (orange) points. The black line is the SVM solution, the red dashed line is Cao et al. (2019)'s solution, and the blue dashed line is ours. Right: Full data distribution showing our solution achieves the lowest generalization error.

**Algorithms of (Cao et al., 2019).** The theoretical analysis of Cao et al. (2019) is limited to the special case of binary classification with linear hypotheses in the separable case. They propose an algorithm based on distinct positive and negative *geometric margins*, justified by their analysis. (Note that our analysis is grounded in the more general notion of *confidence margins* and applies to both separable and non-separable cases, and to general hypothesis sets.)

Their analysis contradicts the recommendations of our theory. Indeed, it is instructive to compare our margin values in the separable case with those derived from the analysis of Cao et al. (2019), in the special case they consider. The margin values proposed in their work are:

$$\rho_+ = \frac{2m_-^{\frac{1}{4}}}{m_+^{\frac{1}{4}} + m_-^{\frac{1}{4}}}\rho_{\text{geom}}, \qquad \rho_- = \frac{2m_+^{\frac{1}{4}}}{m_+^{\frac{1}{4}} + m_-^{\frac{1}{4}}}\rho_{\text{geom}}.$$

Thus, disregarding the suboptimal exponent of $\frac{1}{4}$ compared to $\frac{1}{3}$, which results from a less precise technical analysis, the margin values recommended in their work directly contradict those suggested by our analysis, see Eqn. (2). Specifically, their analysis advocates for a smaller positive margin when $m_+ > m_-$, whereas our theoretical analysis prescribes the opposite. This discrepancy stems from the analysis in (Cao et al., 2019), which focuses on the *balanced loss*, which equalizes the impact of each class by weighting errors inversely to class frequency, and deviates fundamentally from the standard zero-one loss we consider. Figure 1 illustrates these contrasting solutions in a specific case of separable data. On the standard zero-one loss, our approach obtains a lower error.

Although their analysis is restricted to the linearly separable binary case, the authors extend their work to the non-separable multi-class setting by introducing a loss function (LDAM) and algorithm. Their loss function is an instance of the family of logistic loss modifications, with an additive class label-dependent parameter $\Delta_k = C/m_k^{1/4}$ inspired by their analysis in the separable case, where $k$ denotes the label and $C$ a hyperparameter. In the next section, we will compare our proposed algorithm with this technique as well

*Table 1.* Accuracy of ResNet-34 on *long-tailed* imbalanced CIFAR-10, CIFAR-100 and Tiny ImageNet; Means ± standard deviations over five runs for IMMAX and a number of baseline techniques.

| Method | Ratio | CIFAR-10 | CIFAR-100 | Tiny ImageNet |
|---|---|---|---|---|
| CE | | 94.81 ± 0.38 | 78.78 ± 0.49 | 61.72 ± 0.68 |
| RW | | 92.36 ± 0.11 | 67.52 ± 0.76 | 48.16 ± 0.72 |
| BS | | 93.62 ± 0.25 | 72.27 ± 0.73 | 54.18 ± 0.65 |
| EQUAL | | 94.21 ± 0.21 | 76.23 ± 0.80 | 60.63 ± 0.85 |
| LA | 200 | 94.59 ± 0.45 | 78.54 ± 0.49 | 61.83 ± 0.78 |
| CB | | 94.95 ± 0.46 | 79.36 ± 0.81 | 62.51 ± 0.71 |
| FOCAL | | 94.96 ± 0.39 | 79.53 ± 0.75 | 62.70 ± 0.79 |
| LDAM | | 95.45 ± 0.38 | 79.18 ± 0.71 | 63.70 ± 0.62 |
| **IMMAX** | | **96.11 ± 0.34** | **80.47 ± 0.68** | **65.20 ± 0.65** |
| CE | | 95.65 ± 0.23 | 70.05 ± 0.36 | 51.17 ± 0.66 |
| RW | | 93.32 ± 0.51 | 63.35 ± 0.26 | 43.73 ± 0.54 |
| BS | | 94.80 ± 0.26 | 65.36 ± 0.69 | 47.06 ± 0.73 |
| EQUAL | | 95.15 ± 0.39 | 68.81 ± 0.29 | 50.34 ± 0.78 |
| LA | 100 | 95.75 ± 0.17 | 70.19 ± 0.78 | 51.27 ± 0.57 |
| CB | | 95.83 ± 0.11 | 69.85 ± 0.75 | 51.58 ± 0.65 |
| FOCAL | | 95.72 ± 0.11 | 70.33 ± 0.42 | 51.66 ± 0.78 |
| LDAM | | 95.85 ± 0.10 | 70.43 ± 0.52 | 52.00 ± 0.53 |
| **IMMAX** | | **96.56 ± 0.18** | **71.51 ± 0.34** | **53.47 ± 0.72** |
| CE | | 93.05 ± 0.18 | 70.43 ± 0.27 | 53.22 ± 0.42 |
| RW | | 91.45 ± 0.26 | 67.35 ± 0.51 | 48.46 ± 0.78 |
| BS | | 91.84 ± 0.30 | 66.52 ± 0.39 | 51.22 ± 0.53 |
| EQUAL | | 92.30 ± 0.18 | 68.64 ± 0.60 | 51.77 ± 0.30 |
| LA | 10 | 92.84 ± 0.43 | 70.16 ± 0.58 | 53.75 ± 0.20 |
| CB | | 92.96 ± 0.27 | 70.31 ± 0.63 | 53.66 ± 0.58 |
| FOCAL | | 93.09 ± 0.33 | 70.70 ± 0.36 | 53.26 ± 0.50 |
| LDAM | | 93.16 ± 0.25 | 70.94 ± 0.29 | 53.61 ± 0.20 |
| **IMMAX** | | **93.68 ± 0.12** | **71.93 ± 0.36** | **54.89 ± 0.44** |

*Table 2.* Accuracy of ResNet-34 on *step-imbalanced* CIFAR-10, CIFAR-100 and Tiny ImageNet; Means ± standard deviations over five runs for IMMAX and a number of baseline techniques.

| Method | Ratio | CIFAR-10 | CIFAR-100 | Tiny ImageNet |
|---|---|---|---|---|
| CE | | 94.71 ± 0.24 | 77.07 ± 0.55 | 61.61 ± 0.53 |
| RW | | 90.31 ± 0.38 | 72.59 ± 0.26 | 58.49 ± 0.61 |
| BS | | 90.69 ± 0.41 | 74.18 ± 0.62 | 61.11 ± 0.32 |
| EQUAL | | 93.43 ± 0.23 | 76.85 ± 0.38 | 61.81 ± 0.39 |
| LA | 200 | 94.85 ± 0.18 | 76.89 ± 0.74 | 61.51 ± 0.78 |
| CB | | 94.92 ± 0.13 | 77.04 ± 0.13 | 61.55 ± 0.57 |
| FOCAL | | 94.78 ± 0.16 | 77.10 ± 0.62 | 61.77 ± 0.51 |
| LDAM | | 94.85 ± 0.23 | 77.18 ± 0.50 | 62.54 ± 0.51 |
| **IMMAX** | | **95.42 ± 0.30** | **78.21 ± 0.48** | **63.57 ± 0.36** |
| CE | | 95.03 ± 0.21 | 76.92 ± 0.27 | 60.62 ± 0.53 |
| RW | | 90.74 ± 0.19 | 68.17 ± 0.82 | 53.24 ± 0.65 |
| BS | | 93.24 ± 0.36 | 70.97 ± 0.35 | 60.07 ± 0.23 |
| EQUAL | | 94.04 ± 0.30 | 77.17 ± 0.20 | 60.46 ± 0.64 |
| LA | 100 | 94.83 ± 0.11 | 77.27 ± 0.34 | 60.81 ± 0.46 |
| CB | | 95.08 ± 0.28 | 76.88 ± 0.44 | 60.63 ± 0.37 |
| FOCAL | | 95.07 ± 0.34 | 77.00 ± 0.34 | 60.72 ± 0.36 |
| LDAM | | 95.17 ± 0.24 | 77.05 ± 0.45 | 62.33 ± 0.46 |
| **IMMAX** | | **96.05 ± 0.15** | **78.17 ± 0.35** | **63.04 ± 0.60** |
| CE | | 92.95 ± 0.18 | 74.43 ± 0.38 | 59.68 ± 0.29 |
| RW | | 90.64 ± 0.15 | 68.65 ± 0.49 | 46.97 ± 0.73 |
| BS | | 92.55 ± 0.26 | 69.55 ± 0.84 | 56.70 ± 0.34 |
| EQUAL | | 92.62 ± 0.24 | 72.64 ± 0.61 | 60.34 ± 0.52 |
| LA | 10 | 93.55 ± 0.30 | 74.60 ± 0.26 | 60.36 ± 0.28 |
| CB | | 93.54 ± 0.15 | 74.63 ± 0.36 | 59.88 ± 0.29 |
| FOCAL | | 93.11 ± 0.16 | 74.51 ± 0.41 | 59.75 ± 0.44 |
| LDAM | | 93.34 ± 0.16 | 74.82 ± 0.46 | 61.11 ± 0.30 |
| **IMMAX** | | **93.93 ± 0.18** | **75.86 ± 0.26** | **61.93 ± 0.25** |

as a number of other baselines.

# 7. Experiments

In this section, we present experimental results for our IMMAX algorithm, comparing it to baseline methods in minimizing the standard zero-one misclassification loss on CIFAR-10, CIFAR-100 (Krizhevsky, 2009) and Tiny ImageNet (Le & Yang, 2015) datasets. .

Starting with multi-class classification, we strictly followed the experimental setup of Cao et al. (2019), adopting the same training procedure and neural network architectures. Specifically, we used ResNet-34 with ReLU activations (He et al., 2016), where ResNet-$n$ denotes a residual network with $n$ convolutional layers. For CIFAR-10 and CIFAR-100, we applied standard data augmentations, including 4-pixel padding followed by $32 \times 32$ random crops and random horizontal flips. For Tiny ImageNet, we used 8-pixel padding followed by $64 \times 64$ random crops. All models were trained using Stochastic Gradient Descent (SGD) with Nesterov momentum (Nesterov, 1983), a batch size of $1,024$, and a weight decay of $1 \times 10^{-3}$. Training spanned 200 epochs, using a cosine decay learning rate schedule (Loshchilov &

Hutter, 2022) without restarts, with the initial learning rate set to $0.2$. For all the baselines and the IMMAX algorithm, the hyperparameters were selected through cross-validation, see Appendix B for details.

To create imbalanced versions of the datasets, we reduced the percent of examples per class identically in the training and test sets. Following (Cao et al., 2019), we consider two types of imbalances: long-tailed imbalance (Cui et al., 2019) and step imbalance (Buda et al., 2018). The imbalance ratio, $\frac{\max_{k=1}^{c} m_k}{\min_{k=1}^{c} m_k}$, represents the ratio of sample sizes between the most frequent and least frequent classes. In the long-tailed imbalance setting, class sample sizes decrease exponentially across classes. In the step setting, minority classes all have the same sample size, as do the frequent classes, creating a clear distinction between the two groups.

We compare our IMMAX algorithm with widely used baselines, including the cross-entropy (CE) loss, Re-Weighting (RW) method (Xie & Manski, 1989; Morik et al., 1999), Balanced Softmax (BS) loss (Jiawei et al., 2020), Equalization loss (Tan et al., 2020), Logit Adjusted (LA) loss (Menon et al., 2021), Class-Balanced (CB) loss (Cui et al., 2019), the FOCAL loss in (Ross & Dollár, 2017) and the LDAM loss in

(Cao et al., 2019) also detailed in Appendix B. We average accuracies on the imbalanced test set over five runs and report the means and standard deviations. Note that IMMAX is not optimized for other objectives, such as the balanced loss, and thus is not expected to outperform state-of-the-art methods tailored to those metrics.

Table 1 and Table 2 highlight that IMMAX consistently outperforms all baseline methods on both the long-tailed and step-imbalanced datasets across all evaluated imbalance ratios (200, 100, and 10). In every scenario, IMMAX achieves an absolute accuracy improvement of at least 0.6% over the runner-up algorithm. Note, that for the long-tailed distributions, the more imbalanced the dataset is, the more beneficial IMMAX becomes compared to the baselines.

Finally, in Table 3, we include binary classification results on CIFAR-10 obtained by classifying one category, e.g., airplane versus all the others using linear models. Table 3 shows that IMMAX outperforms baselines.

Let us emphasize that our work is based on a novel, principled surrogate loss function designed for imbalanced data. Accordingly, we compare our new loss function directly against existing ones without incorporating additional techniques. However, all these loss functions, including ours, can be combined with existing data modification methods such as oversampling (Chawla et al., 2002) and undersampling (Wallace et al., 2011; Kubat & Matwin, 1997), as well as optimization strategies like the deferred re-balancing schedule proposed in (Cao et al., 2019), to further enhance performance. For a fair comparison of loss functions, we deliberately excluded these techniques from our experiments.

## 8. Discussion

**Balanced accuracy.** Our work focuses on the standard and unmodified zero-one misclassification loss, which remains the primary objective in many machine learning applications, as discussed in Section 1. Accordingly, we report standard accuracy based on this loss function in Section 7. In contrast, some previous studies (e.g., Cao et al. (2019)) report "balanced accuracy", which equalizes the impact of each class by weighting errors inversely to class frequency.

**Selection of $\rho_k$.** As noted in Section 5, while $\rho_k$ can be freely tuned, the search can be guided by the vector $[\rho_k/\rho]_k$ near r, the optimal values in the separable case, a strategy we used in our experiments. For large multi-class settings, we further reduced the search space by assigning shared $\rho_k$ to rare classes and distinct ones to frequent classes, improving practicality with minimal performance impact.

**Balanced data.** IMMAX extends several classical algorithms, such as SVM (Cortes & Vapnik, 1995) and logistic regression (Verhulst, 1838), to the more general setting of

*Table 3.* Accuracy of linear models on binarized version of CIFAR-10; Means ± standard deviations for hinge loss, IMMAX and LDAM.

| Method | Airplane | Automobile | Horse |
|--------|----------|------------|-------|
| HINGE | $90.17 \pm 0.09$ | $91.01 \pm 0.13$ | $90.58 \pm 0.11$ |
| LDAM | $90.37 \pm 0.01$ | $90.44 \pm 0.02$ | $90.17 \pm 0.01$ |
| **IMMAX** | $\mathbf{91.02 \pm 0.06}$ | $\mathbf{91.26 \pm 0.05}$ | $\mathbf{91.03 \pm 0.03}$ |

imbalanced data. When the training dataset is balanced, the theoretically optimal values of $\rho_k$ coincide across all classes, and IMMAX reduces to standard methods. For example, with the logistic loss, it recovers the standard softmax cross-entropy loss with a suitable regularization parameter, thus yielding the same performance.

**Large-scale datasets.** The dependency of our solution on sample size and dimensionality is similar to that of standard neural networks trained with cross-entropy loss (that is the logistic loss when softmax is applied to logits). Thus, our approach remains practical for large-scale datasets when using optimizers such as SGD (Bottou, 2010), AdaGrad (Duchi et al., 2011) or Adam (Kingma & Ba, 2015). Our solution does depend on the number of classes, but this dependency is inherent to standard multi-class neural network solutions as well.

**Contrastive loss and temperature scaling**. The form of our IMMAX loss function has some similarity with supervised contrastive losses (e.g., Khosla et al. (2020)), where a scalar temperature parameter is used in the inner product argument of the exponential. However, in our case, distinct parameters are introduced to allow different confidence margins across classes, serving a different purpose than in contrastive learning. Nevertheless, our margin analysis could provide a useful tool for analyzing contrastive learning. Alternatively, the IMMAX loss can be viewed as a form of class-dependent temperature scaling derived from the logistic loss. In fact, our theoretical framework offers insight into the role of temperature parameters more broadly.

## 9. Conclusion

We introduced a rigorous theoretical framework for addressing class imbalance, culminating in the class-imbalanced margin loss and IMMAX algorithms for binary and multi-class classification. These algorithms are grounded in strong theoretical guarantees, including $\mathcal{H}$-consistency and robust generalization bounds. Empirical results confirm that our algorithms outperform existing methods while remaining aligned with key theoretical principles. Our analysis is not limited to misclassification loss and can be adapted to other objectives like balanced loss, offering broad applicability. We believe these contributions offer a significant step towards principled solutions for class imbalance across a diverse range of machine learning applications.

## Acknowledgements

We thank the anonymous reviewers for their valuable feedback and constructive suggestions.

## Impact Statement

This paper presents work whose goal is to advance the field of Machine Learning. There are many potential societal consequences of our work, none which we feel must be specifically highlighted here.

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

# Contents of Appendix

# A. Related Work

This section provides an expanded discussion of related work on class imbalance in machine learning.

The class imbalance problem, defined by a significant disparity in the number of instances across classes within a dataset, is a common challenge in machine learning applications (Lewis & Gale, 1994; Fawcett & Provost, 1996; Kubat & Matwin, 1997; Kang et al., 2021; Menon et al., 2021; Liu et al., 2019; Cui et al., 2019). This issue is prevalent in many real-world binary classification scenarios, and arguably even more so in multi-class problems with numerous classes. In such cases, a few majority classes often dominate the dataset, leading to a "long-tailed" distribution. Classifiers trained on these imbalanced datasets often struggle, performing similarly to a naive baseline that simply predicts the majority class.

The problem has been widely studied in the literature (Cardie & Nowe, 1997; Kubat & Matwin, 1997; Chawla et al., 2002; He & Garcia, 2009; Wallace et al., 2011). It includes numerous methods including standard Softmax, class-sensitive learning, Weighted Softmax, weighted 0/1 loss (Gabidolla et al., 2024), size-invariant metrics for Imbalanced Multi-object Salient Object Detection studied by Li et al. (2024a) as well as Focal loss (Lin et al., 2017), LDAM (Cao et al., 2019), ESQL (Tan et al., 2020), Balanced Softmax (Jiawei et al., 2020), LADE (Hong et al., 2021)), logit adjustment (UNO-IC (Tian et al., 2020), LSC (Wei et al., 2024)), transfer learning (SSP (Yang & Xu, 2020)), data augmentation (RSG (Wang et al., 2021a), BSGAL (Zhu et al., 2024), ELTA (Liu et al., 2024), OT (Gao et al., 2023)), representation learning (OLTR (Liu et al., 2019), PaCo (Cui et al., 2021), DisA (Gao et al., 2024), RichSem (Meng et al., 2023), RBL (Meng et al., 2023), WCDAS (Han, 2023)), classifier design (De-confound (Tang et al., 2020), (Yang et al., 2022; Kasarla et al., 2022), LIFT (Shi et al., 2024), SimPro (Du et al., 2024)), decoupled training (Decouple-IB-CRT (Kang et al., 2020), CB-CRT (Kang et al., 2020), SR-CRT (Kang et al., 2020), PB-CRT (Kang et al., 2020), MiSLAS (Zhong et al., 2021)), ensemble learning (BBN (Zhou et al., 2020), LFME (Xiang et al., 2020), RIDE (Wang et al., 2021b), ResLT (Cui et al., 2022), SADE (Zhang et al., 2022), DirMixE (Yang et al., 2024)). An interesting recent study characterizes the asymptotic performances of linear classifiers trained on imbalanced datasets for different metrics (Loffredo et al., 2024).

Due to space restrictions, we cannot give a detailed discussion of all these methods. Instead, we will describe and discuss several broad categories of existing methods to tackle this problem and refer to reader to a recent survey of Zhang et al. (2023) for more details. These methods fall into the following broad categories.

**Data modification methods.** These include methods such as oversampling the minority class (Chawla et al., 2002), undersampling the majority class (Wallace et al., 2011; Kubat & Matwin, 1997), or generating synthetic samples (e.g., SMOTE (Chawla et al., 2002; Qiao & Liu, 2008; Han et al., 2005)), aim to rebalance the dataset before training (Chawla et al., 2002; Estabrooks et al., 2004; Liu et al., 2008; Zhang & Pfister, 2021; Shi et al., 2023).

**Cost-sensitive techniques.** These techniques, including cost-sensitive learning and the incorporation of class weights assign different penalization costs to losses on different classes. They include cost-sensitive SVM (Iranmehr et al., 2019; Masnadi-Shirazi & Vasconcelos, 2010) and other cost-senstive methods (Elkan, 2001; Zhou & Liu, 2005; Zhao et al., 2018; Zhang et al., 2018; 2019; Sun et al., 2007; Fan et al., 2017; Jamal et al., 2020; Zhang et al., 2022; Wang et al., 2022; Xiao et al., 2023; Suh & Seo, 2023). The weights are often determined by the relative number of samples in each class or a notion of effective sample size (Cui et al., 2019).

These two method categories are very related and can actually be shown to be equivalent in the limit. Cost-sensitive methods can be viewed as more efficient, flexible and principled techniques for implementing data sampling methods. However, these methods often risk overfitting the minority class or discarding valuable information from the majority class. Both methods inherently bias the input training data distribution and suffer from Bayes-inconsistency (in Section 6, we prove that cost-sensitive methods do not admit Bayes-consistency). While they have been both reported to be effective in various instances, this varies and depends on the problem, the distribution, the choice of predictors, and the performance metric adopted and they have been reported not to be effective in all cases (Van Hulse et al., 2007). Additionally, cost-sensitive methods often resort to careful tuning of hyperparameters. Hybrid approaches attempt to combine the strengths of data modification and cost-sensitive methods but often inherit their respective limitations.

**Logistic loss modifications.** A family of more recent methods rely on logistic loss modifications. They consist of modifying the logistic loss by augmenting each logit (or predicted score) with an additive hyperparameter. They can be equivalently described as a cost-sensitive modification of the exponential terms appearing in the definition of the logistic loss. They include the Balanced Softmax loss (Jiawei et al., 2020), the Equalization loss (Tan et al., 2020), and the LDAM loss (Cao et al., 2019). Other similar additive change methods use quadratically many hyperparameters with a distinct additive parameter for each pair of logits. They include the logit adjustment methods of Menon et al. (2021) and Khan et al. (2019). Menon

et al. (2021) argue that their specific choice of the hyperparameter values is Bayes-consistent. A multiplicative modification of the logits, with one hyperparameter per class label is advocated by Ye et al. (2020). This can be equivalently viewed as normalizing scoring functions (or feature vectors in the linear case) beforehand, which is a standard method used in many learning applications, irrespective of the presence of imbalanced classes. The Vector-Scaling loss of Kini et al. (2021) combines the additive modification of the logits with this multiplicative change. These authors further present an analysis of this method in the case of linear predictors, underscoring the specific benefits of the multiplicative changes. As already pointed out, the multiplicative changes coincide with prior rescaling or renormalization of the feature vectors, however.

**Other methods.** Additional approaches for tackling imbalanced datasets (see Zhang et al. (2023)) include post-hoc correction of decision thresholds (Fawcett & Provost, 1996; Collell et al., 2016) or weights (Kang et al., 2020; Kim & Kim, 2020)], as well as information and data augmentation via transfer learning, or distillation (Li et al., 2024b).

Despite significant advances, these techniques face persistent challenges.

First, most existing solutions are heuristic-driven and lack a solid theoretical foundation, making their performance difficult to predict across varying contexts.

In fact, we are not aware of any analysis of the generalization guarantees for these methods, with the exception of that of (Cao et al., 2019; Jiawei et al., 2020; Wang et al., 2023). However, as further discussed in Section 6, the analysis presented by these authors is limited to the *balanced loss*, which equalizes the impact of each class by weighting errors inversely to class frequency. More specifically, the analysis in (Cao et al., 2019) is limited to binary classification and holds only for the separable case. The balanced loss function differs from the target misclassification loss. It has been argued, and that is important, that the balanced loss admits beneficial fairness properties when class labels correlate with demographic attributes as it treats all class errors equally. The balanced loss is also the metric considered in the analysis of several of the logistic loss modifications papers (Cao et al., 2019; Menon et al., 2021; Ye et al., 2020; Kini et al., 2021). However, class labels do not alway relate to demographic attributes. Furthermore, many other criteria are considered for fairness purposes and in many machine learning applications, the misclassification remains the key target loss function to minimize. We will show that, even in the special case of the analysis of (Cao et al., 2019), the solution they propose is the opposite of the one corresponding to our theoretical analysis for the standard misclassification loss. We further show that their solution is empirically outperformed by ours.

Second, the evaluation of these methods is frequently biased toward alternative metrics such as F1-measure, AUC, or other metrics weighting false or true positive rate differently, which may obscure their true effectiveness on standard misclassification. Additionally, these methods often seem to struggle with extreme imbalances or when the minority class exhibits high intra-class variability.

We refer to Zhang et al. (2023) for more details about work related to learning from imbalanced data.

## B. Experimental details

In this section, we provide further experimental details. We first discuss the loss functions for the baselines and then provide ranges of hyperparameters tested via cross-validation.

**Baseline algorithms.** In Section 7, we compared our IMMAX algorithm with well-known baselines, including the cross-entropy (CE) loss, Re-Weighting (RW) method (Xie & Manski, 1989; Morik et al., 1999), Balanced Softmax (BS) loss (Jiawei et al., 2020), Equalization loss (Tan et al., 2020), Logit Adjusted (LA) loss (Menon et al., 2021), Class-Balanced (CB) loss (Cui et al., 2019), the FOCAL loss in (Ross & Dollár, 2017) and the LDAM loss in (Cao et al., 2019).

The IMMAX algorithm optimizes the loss function:

$$\forall (h, x, y), \quad \mathsf{L}_{\text{IMMAX}}(h, x, y) = \log\left(\sum_{j=1}^{c} e^{\frac{h(x,j)-h(x,y)}{\rho_y}}\right),$$

where $\rho_k > 0$ for $k \in [c]$ are hyperparameters. In comparison, the baselines optimize the following loss functions:

- Cross-entropy (CE) loss:

$$\forall (h, x, y), \quad \mathsf{L}_{\text{CE}}(h, x, y) = -\log\left(\frac{e^{h(x,y)}}{\sum_{j=1}^{c} e^{h(x,j)}}\right).$$

- Re-Weighting (RW) method (Xie & Manski, 1989; Morik et al., 1999): Each sample is re-weighted by the inverse of its class's sample size and subsequently normalized such that the average weight within each mini-batch is 1. This is equivalent to minimizing the loss function given below:

$$\forall (h, x, y), \quad \mathsf{L}_{\mathrm{RW}}(h, x, y) = -\frac{m}{m_y} \log\left( \frac{e^{h(x,y)}}{\sum_{j=1}^{c} e^{h(x,j)}} \right).$$

- Balanced Softmax (BS) loss (Jiawei et al., 2020):

$$\forall (h, x, y), \quad \mathsf{L}_{\mathrm{BS}}(h, x, y) = -\log\left( \frac{m_y e^{h(x,y)}}{\sum_{j=1}^{c} m_j e^{h(x,j)}} \right).$$

- Equalization loss (Tan et al., 2020):

$$\forall (h, x, y), \quad \mathsf{L}_{\mathrm{EQUAL}}(h, x, y) = -\log\left( \frac{e^{h(x,y)}}{\sum_{j=1}^{c} w_j e^{h(x,j)}} \right),$$

with the weight $w_j$ computed by $w_j = 1 - \beta 1_{\frac{m_j}{m} < \lambda} 1_{y \neq j}$, where $\beta \sim \text{Bernoulli}(p)$ is a Bernoulli distribution. Here, $1 > p > 0$ and $1 > \lambda > 0$ are two hyperparameters.

- Logit Adjusted (LA) loss (Menon et al., 2021):

$$\forall (h, x, y), \quad \mathsf{L}_{\mathrm{LA}}(h, x, y) = -\log\left( \frac{e^{h(x,y) + \tau \log(m_y)}}{\sum_{j=1}^{c} e^{h(x,j) + \tau \log(m_j)}} \right),$$

where $\tau > 0$ is a hyperparameter.

- Class-Balanced (CB) loss (Cui et al., 2019):

$$\forall (h, x, y), \quad \mathsf{L}_{\mathrm{CB}}(h, x, y) = -\frac{1 - \gamma}{1 - \gamma^{\frac{m_y}{m}}} \log\left( \frac{e^{h(x,y)}}{\sum_{j=1}^{c} e^{h(x,j)}} \right),$$

where $1 > \gamma > 0$ is a hyperparameter.

- FOCAL loss in (Ross & Dollár, 2017):

$$\forall (h, x, y), \quad \mathsf{L}_{\mathrm{FOCAL}}(h, x, y) = -\left( 1 - \frac{e^{h(x,y)}}{\sum_{j=1}^{c} e^{h(x,j)}} \right)^{\gamma} \log\left( \frac{e^{h(x,y)}}{\sum_{j=1}^{c} e^{h(x,j)}} \right),$$

where $\gamma \geq 0$ is a hyperparameter.

- LDAM loss in (Cao et al., 2019):

$$\forall (h, x, y), \quad \mathsf{L}_{\mathrm{LDAM}}(h, x, y) = -\log\left( \frac{e^{h(x,y) - \Delta_y}}{e^{h(x,y) - \Delta_y} + \sum_{j \neq y} e^{h(x,j)}} \right),$$

where $\Delta_j = \frac{C}{m_j^{\frac{1}{4}}}$ for $j \in [c]$ and $C > 0$ is a hyperparameter.

**Discussion.** Among these baselines, RW method, CB loss, and FOCAL loss are cost-sensitive methods, while BS loss, EQUAL loss, LA loss, and LDAM loss are logistic loss modification methods. Note that when $\tau = 1$, the LA loss is the same as the BS loss; when $\tau = 0$, the FOCAL loss is the same as the CE loss. Also note that in the balanced setting where $m_j = m/c$ for $j \in [c]$, the RW method, BS loss, LA loss and CB loss are the same as the CE loss.

**Hyperparameter search.** As mentioned in Section 7, all hyperparameters were selected through cross-validation for all the baselines and the IMMAX algorithm. More specifically, the parameter ranges for each method are as follows. Note that the CE loss, RW method and BS loss do not have any hyperparameters.

- EQUAL loss: following (Tan et al., 2020), $p$ is chosen from $\{0.1, 0.2, 0.3, 0.4, 0.5, 0.6, 0.7, 0.8, 0.9\}$ and $\lambda$ is chosen from $\{0.176, 0.5, 0.8, 1.5, 1.76, 2.0, 3.0, 5.0\} \times 10^{-3}$.

- LA loss: following (Menon et al., 2021), $\tau$ is chosen from $\{0.1, 0.2, 0.3, 0.4, 0.5, 0.6, 0.7, 0.8, 0.9, 1.0\}$ and $\{1.5, 2.0, 2.5, 3.0, 3.5, 4.0, 4.5, 5.0, 5.5, 6.0, 6.5, 7.0, 7.5, 8.0, 8.5, 9.0, 9.5, 10.0\}$. When $\tau = 1$ (the suggested value in (Menon et al., 2021)), the LA loss is equivalent to the BS loss. We observed improved performance for small values of $\tau < 1$ when minimizing the standard zero-one misclassification loss. Therefore, we conducted a finer search between 0 and 1.

- CB loss: following (Cui et al., 2019), $\gamma$ is chosen from $\{0.1, 0.2, 0.3, 0.4, 0.5, 0.6, 0.7, 0.8, 0.9, 0.99, 0.999, 0.9999\}$. While the default values of $\{0.9, 0.99, 0.999, 0.9999\}$ are suggested in (Cui et al., 2019), we observed that they are not effective for minimizing the standard zero-one misclassification loss. We found that performance is typically better when $\gamma$ is close to 0.

- FOCAL loss: $\gamma$ is chosen from $\{1.0, 1.5, 2.0, 2.5, 3.0, 3.5, 4.0, 4.5, 5.0, 5.5, 6.0, 6.5, 7.0, 7.5, 8.0, 8.5, 9.0, 9.5, 10.0\}$ and $\{0.0, 0.1, 0.2, 0.3, 0.4, 0.5, 0.6, 0.7, 0.8, 0.9\}$ following (Ross & Dollár, 2017). We observe that performance is typically better when $\gamma$ is less than 1. Therefore, we conducted a finer search between 0 and 1.

- LDAM loss: following (Cao et al., 2019), $C$ is chosen from $\{10^{-4}, 10^{-3}, 10^{-2}, 10^{-1}, 1.0, 10.0, 100.0, 1000.0, 10000.0\}$ and $\{5 \times 10^{-4}, 5 \times 10^{-3}, 5 \times 10^{-2}, 5 \times 10^{-1}, 5.0, 50.0, 500.0, 5000.0\}$.

- IMMAX loss: following Section 4 and Appendix F.4, each $\rho_k$ is searched within a symmetric interval centered at the theoretically optimal value $\rho_k^* = \dfrac{m_k^{\frac{1}{3}}}{\Sigma_{j \in [c]} m_j^{\frac{1}{3}}}$, with width $\delta_k$; that is, over the interval $[\rho_k^* - \delta_k, \rho_k^* + \delta_k]$, where $\delta_k$ is chosen relative to the scale of $\rho_k^*$. In particular, we set $\delta_k \approx 0.8\rho_k^*$ and evaluate approximately 10 values within each interval. Empirically, performance is not sensitive to variations within this neighborhood. In the step imbalanced setting, we assign identical $\rho_k$ values to minority classes and distinct $\rho_k$ values to frequent classes before the search.

## C. Proof of Theorem 6.1

**Theorem 6.1** (Negative results for resampling and cost-sensitive methods). *If $c_+ \neq c_-$, then $\mathsf{L}_{c_+,c_-}$ is not Bayes-consistent with respect to $\ell_{0-1}$.*

*Proof.* Consider a singleton distribution concentrated at a point $x$. Without loss of generality, assume that $c_+ > c_- > 0$. Next, consider the conditional distribution $\eta(x) = \mathbb{P}[Y = +1 \mid X = x]$ denote the conditional probability that $Y = +1$ given $X = x$ with $\eta(x) = \frac{1}{2} - \epsilon$, for $\epsilon \in (0, \frac{1}{2})$. By the proof of Theorem 3.3, the best-in-class error for the zero-one loss can be expressed as follows:

$$\inf_{h \in \mathcal{H}} \mathcal{R}_{\ell_{0-1}}(h) = \eta(x),$$

which can be achieved by any $h_{\ell_{0-1}}^*$ such that $h_{\ell_{0-1}}^*(x) < 0$, that is a hypothesis *all-negative* on $x$. For the cost-sensitive loss function $\mathsf{L}_{c_+,c_-}$, the generalization error can be expressed as follows:

$$\mathcal{R}_{\mathsf{L}_{c_+,c_-}}(h) = \eta(x)c_+ \mathbb{1}_{h(x)<0} + (1 - \eta(x))c_- \mathbb{1}_{h(x)\geq 0}.$$

Thus, for any $c_+ > c_- > 0$, there exists $\epsilon \in (0, \frac{1}{2})$ such that the following holds:

$$(1 - \eta(x))c_- < \eta(x)c_+ \iff \frac{\frac{1}{2} + \epsilon}{\frac{1}{2} - \epsilon} < \frac{c_+}{c_-}$$

$$\iff 0 < \epsilon < \frac{\frac{1}{2}c_+ - \frac{1}{2}c_-}{c_+ + c_-} < \frac{1}{2},$$

where we used the fact that $x \mapsto (1 - x)/x = 1/x - 1$ is a bijection from $(0, 1]$ to $[0, +\infty)$. For this $\epsilon$, the best-in-class error of $\mathsf{L}_{c_+,c_-}$ is

$$\inf_{h \in \mathcal{H}} \mathcal{R}_{\mathsf{L}_{c_+,c_-}}(h) = (1 - \eta(x))c_-,$$

which can be achieved by any *all-positive* $h_{\mathsf{L}_{c_+,c_-}}^*$ such that $h_{\mathsf{L}_{c_+,c_-}}^*(x) \geq 0$. Thus, $h_{\mathsf{L}_{c_+,c_-}}^*$ differs from $h_{\ell_{0-1}}^*$, which implies that $\mathsf{L}_{c_+,c_-}$ is not Bayes-consistent with respect to $\ell_{0-1}$. $\qquad\square$

# D. Binary Classification: Proofs

### D.1. Proof of Lemma 3.2

**Lemma 3.2.** *The class-imbalanced $(\rho_+, \rho_-)$-margin loss function can be equivalently expressed as follows:*

$$\mathsf{L}_{\rho_+,\rho_-}(h,x,y) = \Phi_{\rho_+}(yh(x))1_{h(x)\geq 0} + \Phi_{\rho_-}(yh(x))1_{h(x)<0}.$$

*Proof.* When $yh(x) \leq 0$, we have $\Phi_{\rho_+}(yh(x)) = \Phi_{\rho_-}(yh(x)) = 1$, so the equality holds. When $yh(x) > 0$, we have $y > 0 \iff h(x) > 0$ and $y < 0 \iff h(x) < 0$, which also implies the equality. □

### D.2. Proof of Theorem 3.3

**Theorem 3.3** ($\mathcal{H}$-consistency bound for class-imbalanced margin loss)**.** *Let $\mathcal{H}$ be a complete hypothesis set. Then, for all $h \in \mathcal{H}$, $\rho_+ > 0$, and $\rho_- > 0$, the following bound holds:*

$$\mathcal{R}_{\ell_{0-1}}(h) - \mathcal{R}^*_{\ell_{0-1}}(\mathcal{H}) + \mathcal{M}_{\ell_{0-1}}(\mathcal{H}) \leq \mathcal{R}_{\mathsf{L}_{\rho_+,\rho_-}}(h) - \mathcal{R}^*_{\mathsf{L}_{\rho_+,\rho_-}}(\mathcal{H}) + \mathcal{M}_{\mathsf{L}_{\rho_+,\rho_-}}(\mathcal{H}). \tag{1}$$

*Proof.* Let $\eta(x) = \mathbb{P}[Y = +1 \mid X = x]$ denote the conditional probability that $Y = +1$ given $X = x$. Without loss of generality, assume $\eta(x) \in [0, \frac{1}{2}]$. Then, the conditional error and the best-in-class conditional error of the zero-one loss can be expressed as follows:

$$\mathbb{E}_y[\ell_{0-1}(h,x,y) \mid x] = \eta(x)1_{h(x)<0} + (1 - \eta(x))1_{h(x)\geq 0}$$

$$\inf_{h\in\mathcal{H}} \mathbb{E}_y[\ell_{0-1}(h,x,y) \mid x] = \min\{\eta(x), 1 - \eta(x)\} = \eta(x).$$

Furthermore, the difference between the two terms is given by:

$$\mathbb{E}_y[\ell_{0-1}(h,x,y) \mid x] - \inf_{h\in\mathcal{H}} \mathbb{E}_y[\ell_{0-1}(h,x,y) \mid x] = \begin{cases} 1 - 2\eta(x) & h(x) \geq 0 \\ 0 & h(x) < 0 \end{cases}$$

For the class-imbalanced margin loss, the conditional error can be expressed as follows:

$$\mathbb{E}_y[\mathsf{L}_{\rho_+,\rho_-}(h,x,y) \mid x] = \eta(x)\Phi_{\rho_+}(h(x)) + (1 - \eta(x))\Phi_{\rho_-}(-h(x))$$

$$= \eta(x)\min\left(1, \max\left(0, 1 - \frac{h(x)}{\rho_+}\right)\right) + (1 - \eta(x))\min\left(1, \max\left(0, 1 + \frac{h(x)}{\rho_-}\right)\right)$$

$$= \begin{cases} 1 - \eta(x) & h(x) \geq \rho_+ \\ \eta(x)\left(1 - \frac{h(x)}{\rho_+}\right) + (1 - \eta(x)) & \rho_+ > h(x) \geq 0 \\ \eta(x) + (1 - \eta(x))\left(1 + \frac{h(x)}{\rho_-}\right) & -\rho_- \leq h(x) < 0 \\ \eta(x) & h(x) < -\rho_-. \end{cases}$$

Thus, the best-in-class conditional error can be expressed as follows:

$$\inf_{h\in\mathcal{H}} \mathbb{E}_y[\mathsf{L}_{\rho_+,\rho_-}(h,x,y) \mid x] = \min\{\eta(x), 1 - \eta(x)\} = \eta(x)$$

Consider the case where $h(x) \geq 0$. The difference between the two terms is given by:

$$\mathbb{E}_y[\mathsf{L}_{\rho_+,\rho_-}(h,x,y) \mid x] - \inf_{h\in\mathcal{H}} \mathbb{E}_y[\mathsf{L}_{\rho_+,\rho_-}(h,x,y) \mid x] = \begin{cases} 1 - 2\eta(x) & h(x) \geq \rho_+ \\ \eta(x)\left(1 - \frac{h(x)}{\rho_+}\right) + 1 - 2\eta(x) & \rho_+ > h(x) \geq 0 \end{cases}$$

$$\geq 1 - 2\eta(x)$$

$$= \mathbb{E}_y[\ell_{0-1}(h,x,y) \mid x] - \inf_{h\in\mathcal{H}} \mathbb{E}_y[\ell_{0-1}(h,x,y) \mid x].$$

By taking the expectation of both sides, we obtain:

$$\mathcal{R}_{\ell_{0-1}}(h) - \mathcal{R}^*_{\ell_{0-1}}(\mathcal{H}) + \mathcal{M}_{\ell_{0-1}}(\mathcal{H}) \leq \mathcal{R}_{\mathsf{L}_{\rho_+,\rho_-}}(h) - \mathcal{R}^*_{\mathsf{L}_{\rho_+,\rho_-}}(\mathcal{H}) + \mathcal{M}_{\mathsf{L}_{\rho_+,\rho_-}}(\mathcal{H}),$$

which completes the proof. □

## D.3. Proof of Theorem 3.5

**Theorem 3.5** (Margin bound for imbalanced binary classification). *Let $\mathcal{H}$ be a set of real-valued functions. Fix $\rho_+ > 0$ and $\rho_- > 0$, then, for any $\delta > 0$, with probability at least $1 - \delta$, each of the following holds for all $h \in \mathcal{H}$:*

$$\mathcal{R}_{\ell_{0-1}}(h) \le \widehat{\mathcal{R}}_S^{\rho_+,\rho_-}(h) + 2\mathfrak{R}_m^{\rho_+,\rho_-}(\mathcal{H}) + \sqrt{\frac{\log \frac{1}{\delta}}{2m}}$$

$$\mathcal{R}_{\ell_{0-1}}(h) \le \widehat{\mathcal{R}}_S^{\rho_+,\rho_-}(h) + 2\widehat{\mathfrak{R}}_S^{\rho_+,\rho_-}(\mathcal{H}) + 3\sqrt{\frac{\log \frac{2}{\delta}}{2m}}.$$

*Proof.* Consider the family of functions taking values in $[0, 1]$:

$$\widetilde{\mathcal{H}} = \{z = (x, y) \mapsto \mathsf{L}_{\rho_+,\rho_-}(h, x, y) : h \in \mathcal{H}\}.$$

By (Mohri et al., 2018, Theorem 3.3), with probability at least $1 - \delta$, for all $g \in \widetilde{\mathcal{H}}$,

$$\mathbb{E}[g(z)] \le \frac{1}{m} \sum_{i=1}^m g(z_i) + 2\mathfrak{R}_m(\widetilde{\mathcal{H}}) + \sqrt{\frac{\log \frac{1}{\delta}}{2m}},$$

and thus, for all $h \in \mathcal{H}$,

$$\mathbb{E}[\mathsf{L}_{\rho_+,\rho_-}(h, x, y)] \le \widehat{\mathcal{R}}_S^{\rho_+,\rho_-}(h) + 2\mathfrak{R}_m(\widetilde{\mathcal{H}}) + \sqrt{\frac{\log \frac{1}{\delta}}{2m}}.$$

Since $\mathcal{R}_{\ell_{0-1}}(h) \le \mathcal{R}_{\mathsf{L}_{\rho_+,\rho_-}}(h) = \mathbb{E}[\mathsf{L}_{\rho_+,\rho_-}(h, x, y)]$, we have

$$\mathcal{R}_{\ell_{0-1}}(h) \le \widehat{\mathcal{R}}_S^{\rho_+,\rho_-}(h) + 2\mathfrak{R}_m(\widetilde{\mathcal{H}}) + \sqrt{\frac{\log \frac{1}{\delta}}{2m}}.$$

Since $\Phi_\rho$ is $\frac{1}{\rho}$-Lipschitz, by (Mohri et al., 2018, Lemma 5.7), $\mathfrak{R}_m(\widetilde{\mathcal{H}})$ can be rewritten as follows:

$$
\begin{aligned}
\mathfrak{R}_m(\widetilde{\mathcal{H}}) &= \frac{1}{m} \mathop{\mathbb{E}}_{S,\sigma}\left[\sup_{h \in \mathcal{H}} \sum_{i=1}^m \sigma_i \mathsf{L}_{\rho_+,\rho_-}(h, x_i, y_i)\right] \\
&= \frac{1}{m} \mathop{\mathbb{E}}_{S,\sigma}\left[\sup_{h \in \mathcal{H}} \sum_{i=1}^m \sigma_i[\Phi_{\rho_+}(y_i h(x_i))1_{y_i=+1} + \Phi_{\rho_-}(y_i h(x_i))1_{y_i=-1}]\right] \\
&\le \frac{1}{m} \mathop{\mathbb{E}}_{S,\sigma}\left[\sup_{h \in \mathcal{H}}\left\{\frac{1}{\rho_+}\left(\sum_{i \in I_+} \sigma_i h(x_i)\right) + \frac{1}{\rho_-}\left(\sum_{i \in I_-} -\sigma_i h(x_i)\right)\right\}\right] \\
&= \mathfrak{R}_m^{\rho_+,\rho_-}(\mathcal{H}),
\end{aligned}
$$

where the last equality stems from the fact that the variables $\sigma_i$ and $-\sigma_i$ are distributed in the same way. This proves the first inequality. The second inequality, can be derived in the same way by using the second inequality of (Mohri et al., 2018, Theorem 3.3). $\square$

## D.4. Uniform Margin Bound for Imbalanced Binary Classification

**Theorem D.1** (Uniform margin bound for imbalanced binary classification). *Let $\mathcal{H}$ be a set of real-valued functions. Fix $r_+ > 0$ and $r_- > 0$. Then, for any $\delta > 0$, with probability at least $1 - \delta$, each of the following holds for all $h \in \mathcal{H}$, $\rho_+ \in (0, r_+]$ and $\rho_- \in (0, r_-]$:*

$$\mathcal{R}_{\ell_{0-1}}(h) \le \widehat{\mathcal{R}}_S^{\rho_+,\rho_-}(h) + 4\mathfrak{R}_m^{\rho_+,\rho_-}(\mathcal{H}) + \sqrt{\frac{\log \log_2 \frac{2r_+}{\rho_+}}{m}} + \sqrt{\frac{\log \log_2 \frac{2r_-}{\rho_-}}{m}} + \sqrt{\frac{\log \frac{4}{\delta}}{2m}}$$

$$\mathcal{R}_{\ell_{0-1}}(h) \le \widehat{\mathcal{R}}_S^{\rho_+,\rho_-}(h) + 4\widehat{\mathfrak{R}}_S^{\rho_+,\rho_-}(\mathcal{H}) + \sqrt{\frac{\log \log_2 \frac{2r_+}{\rho_+}}{m}} + \sqrt{\frac{\log \log_2 \frac{2r_-}{\rho_-}}{m}} + 3\sqrt{\frac{\log \frac{8}{\delta}}{2m}}.$$

*Proof.* First, consider two sequences $\left(\rho_+^k\right)_{k\geq 1}$ and $(\epsilon_k)_{k\geq 1}$, with $\epsilon_k \in (0,1]$. By Theorem 3.5, for any fixed $k \geq 1$ and $\rho_- > 0$,

$$\mathbb{P}\left[\sup_{h\in\mathcal{H}} \mathcal{R}_{\ell_{0-1}}(h) - \widehat{\mathcal{R}}_S^{\rho_+^k,\rho_-}(h) > 2\mathfrak{R}_m^{\rho_+^k,\rho_-}(\mathcal{H}) + \epsilon_k\right] \leq e^{-2m\epsilon_k^2}.$$

Choosing $\epsilon_k = \epsilon + \sqrt{\frac{\log k}{m}}$, then, by the union bound, the following holds for any fixed $\rho_- > 0$:

$$\mathbb{P}\left[\sup_{\substack{h\in\mathcal{H}\\k\geq 1}} \mathcal{R}_{\ell_{0-1}}(h) - \widehat{\mathcal{R}}_S^{\rho_+^k,\rho_-}(h) - 2\mathfrak{R}_m^{\rho_+^k,\rho_-}(\mathcal{H}) - \epsilon_k > 0\right]$$

$$\leq \sum_{k\geq 1} e^{-2m\epsilon_k^2} = \sum_{k\geq 1} \exp^{-2m\left(\epsilon+\sqrt{\frac{\log k}{m}}\right)^2} \leq \sum_{k\geq 1} e^{-2m\epsilon^2} e^{-2\log k} = \left(\sum_{k\geq 1} 1/k^2\right)e^{-2m\epsilon^2} \leq 2e^{-2m\epsilon^2}.$$

We can choose $\rho_+^k = r_+/2^k$. For any $\rho_+ \in (0, r_+]$, there exists $k \geq 1$ such that $\rho_+ \in (\rho_+^k, \rho_+^{k-1}]$, with $\rho_+^0 = r_+$. For that $k$, $\rho_+ \leq \rho_+^{k-1} = 2\rho_+^k$, thus $1/\rho_+^k \leq 2/\rho_+$ and $\sqrt{\log k} = \sqrt{\log\log_2(r_+/\rho_+^k)} \leq \sqrt{\log\log_2(2r_+/\rho_+)}$. Furthermore, for any $h \in \mathcal{H}$ and $\rho_- > 0$, $\widehat{\mathcal{R}}_S^{\rho_+^k,\rho_-}(h) \leq \widehat{\mathcal{R}}_S^{\rho_+,\rho_-}(h)$. Thus, the following inequality holds for any fixed $\rho_- > 0$:

$$\mathbb{P}\left[\sup_{\substack{h\in\mathcal{H}\\\rho_+\in(0,r_+]}} \mathcal{R}_{\ell_{0-1}}(h) - \widehat{\mathcal{R}}_S^{\rho_+,\rho_-}(h) - 2\mathfrak{R}_m^{\rho_+/2,\rho_-}(\mathcal{H}) - \sqrt{\frac{\log\log_2(2r_+/\rho_+)}{m}} - \epsilon > 0\right] \leq 2e^{-2m\epsilon^2}. \tag{7}$$

Next, consider two sequences $\left(\rho_-^l\right)_{l\geq 1}$ and $(\epsilon_l)_{l\geq 1}$, with $\epsilon_l \in (0,1]$. By inequality (7), for any fixed $l \geq 1$,

$$\mathbb{P}\left[\sup_{\substack{h\in\mathcal{H}\\\rho_+\in(0,r_+]}} \mathcal{R}_{\ell_{0-1}}(h) - \widehat{\mathcal{R}}_S^{\rho_+,\rho_-^l}(h) - 2\mathfrak{R}_m^{\rho_+/2,\rho_-^l}(\mathcal{H}) - \sqrt{\frac{\log\log_2(2r_+/\rho_+)}{m}} - \epsilon_l > 0\right] \leq 2e^{-2m\epsilon_l^2}.$$

Choosing $\epsilon_l = \epsilon + \sqrt{\frac{\log l}{m}}$, then, by the union bound, the following holds:

$$\mathbb{P}\left[\sup_{\substack{h\in\mathcal{H}\\\rho_+\in(0,r_+]\\l\geq 1}} \mathcal{R}_{\ell_{0-1}}(h) - \widehat{\mathcal{R}}_S^{\rho_+,\rho_-^l}(h) - 2\mathfrak{R}_m^{\rho_+/2,\rho_-^l}(\mathcal{H}) - \sqrt{\frac{\log\log_2(2r_+/\rho_+)}{m}} - \epsilon_l > 0\right]$$

$$\leq \sum_{l\geq 1} 2e^{-2m\epsilon_l^2} = 2\sum_{l\geq 1} \exp^{-2m\left(\epsilon+\sqrt{\frac{\log l}{m}}\right)^2} \leq 2\sum_{l\geq 1} e^{-2m\epsilon^2} e^{-2\log l} = 2\left(\sum_{l\geq 1} 1/l^2\right)e^{-2m\epsilon^2} \leq 4e^{-2m\epsilon^2}.$$

We can choose $\rho_-^l = r_-/2^l$. For any $\rho_- \in (0, r_-]$, there exists $l \geq 1$ such that $\rho_- \in (\rho_-^l, \rho_-^{l-1}]$, with $\rho_-^0 = r_-$. For that $l$, $\rho_- \leq \rho_-^{l-1} = 2\rho_-^l$, thus $1/\rho_-^l \leq 2/\rho_-$ and $\sqrt{\log l} = \sqrt{\log\log_2(r_-/\rho_-^l)} \leq \sqrt{\log\log_2(2r_-/\rho_-)}$. Furthermore, for any $h \in \mathcal{H}$, $\widehat{\mathcal{R}}_S^{\rho_+,\rho_-^l}(h) \leq \widehat{\mathcal{R}}_S^{\rho_+,\rho_-}(h)$. Thus, the following inequality holds:

$$\mathbb{P}\left[\sup_{\substack{h\in\mathcal{H}\\\rho_+\in(0,r_+]\\\rho_-\in(0,r_-]}} \mathcal{R}_{\ell_{0-1}}(h) - \widehat{\mathcal{R}}_S^{\rho_+,\rho_-}(h) - 4\mathfrak{R}_m^{\rho_+,\rho_-}(\mathcal{H}) - \sqrt{\frac{\log\log_2(2r_+/\rho_+)}{m}} - \sqrt{\frac{\log\log_2(2r_-/\rho_-)}{m}} - \epsilon > 0\right] \leq 4e^{-2m\epsilon^2},$$

where we used the fact that $\mathfrak{R}_m^{\rho_+/2,\rho_-/2}(\mathcal{H}) = 2\mathfrak{R}_m^{\rho_+,\rho_-}(\mathcal{H})$. This proves the first statement. The second statement can be proven in a similar way. $\square$

## D.5. Linear Hypotheses

Combining Theorem 4.1 and Theorem 3.5 gives directly the following general margin bound for linear hypotheses with bounded weighted vectors.

**Corollary D.2.** *Let $\mathcal{H} = \{x \mapsto w \cdot x \colon \|w\| \le \Lambda\}$ and assume $\mathcal{X} \subseteq \{x \colon \|x\| \le r\}$. Let $r_+ = \sup_{i \in I_+} \|x_i\|$ and $r_- = \sup_{i \in I_-} \|x_i\|$. Fix $\rho_+ > 0$ and $\rho_- > 0$, then, for any $\delta > 0$, with probability at least $1 - \delta$ over the choice of a sample $S$ of size $m$, the following holds for any $h \in \mathcal{H}$:*

$$\mathcal{R}_{\ell_{0-1}}(h) \le \widehat{\mathcal{R}}_S^{\rho_+, \rho_-}(h) + \frac{2\Lambda}{m}\sqrt{\frac{m_+ r_+^2}{\rho_+^2} + \frac{m_- r_-^2}{\rho_-^2}} + 3\sqrt{\frac{\log \frac{2}{\delta}}{2m}}$$

$$\le \widehat{\mathcal{R}}_S^{\rho_+, \rho_-}(h) + \frac{2\Lambda r}{m}\sqrt{\frac{m_+}{\rho_+^2} + \frac{m_-}{\rho_-^2}} + 3\sqrt{\frac{\log \frac{2}{\delta}}{2m}}.$$

Choosing $\Lambda = 1$, by the generalization of Corollary D.2 to a uniform bound over $\rho_+ \in (0, r_+]$ and $\rho_- \in (0, r_-]$, for any $\delta > 0$, with probability at least $1 - \delta$, the following holds for all $h \in \{x \mapsto w \cdot x \colon \|w\| \le 1\}$, $\rho_+ \in (0, r_+]$ and $\rho_- \in (0, r_-]$:

$$\mathcal{R}_{\ell_{0-1}}(h) \le \widehat{\mathcal{R}}_S^{\rho_+, \rho_-}(h) + \frac{4r}{m}\sqrt{\frac{m_+}{\rho_+^2} + \frac{m_-}{\rho_-^2}} + \sqrt{\frac{\log \log_2 \frac{2r_+}{\rho_+}}{m}} + \sqrt{\frac{\log \log_2 \frac{2r_-}{\rho_-}}{m}} + 3\sqrt{\frac{\log \frac{8}{\delta}}{2m}}. \tag{8}$$

Now, for any $\rho > 0$, the $\rho$-margin loss function is upper bounded by the $\rho$-hinge loss:

$$\forall u \in \mathbb{R}, \quad \Phi_\rho(u) = \min\left(1, \max\left(0, 1 - \frac{u}{\rho}\right)\right) \le \max\left(0, 1 - \frac{u}{\rho}\right).$$

Thus, with probability at least $1 - \delta$, the following holds for all $h \in \{x \mapsto w \cdot x \colon \|w\| \le 1\}$, $\rho_+ \in (0, r_+]$ and $\rho_- \in (0, r_-]$:

$$\begin{aligned}\mathcal{R}_{\ell_{0-1}}(h) \le &\frac{1}{m}\left[\sum_{i \in I_+} \max\left(0, 1 - \frac{y_i h(x_i)}{\rho_+}\right) + \sum_{i \in I_-} \max\left(0, 1 - \frac{y_i h(x_i)}{\rho_-}\right)\right] \\ &+ \frac{4r}{m}\sqrt{\frac{m_+}{\rho_+^2} + \frac{m_-}{\rho_-^2}} + \sqrt{\frac{\log \log_2 \frac{2r_+}{\rho_+}}{m}} + \sqrt{\frac{\log \log_2 \frac{2r_-}{\rho_-}}{m}} + \sqrt{\frac{\log \frac{4}{\delta}}{2m}}.\end{aligned} \tag{9}$$

Since for any $\rho > 0$, $h/\rho$ admits the same generalization error as $h$, with probability at least $1 - \delta$, the following holds for all $h \in \left\{x \mapsto w \cdot x \colon \|w\| \le \frac{1}{\rho_+ + \rho_-}\right\}$, $\rho_+ \in (0, r_+]$ and $\rho_- \in (0, r_-]$:

$$\begin{aligned}\mathcal{R}_{\ell_{0-1}}(h) \le &\frac{1}{m}\left[\sum_{i \in I_+} \max\left(0, 1 - y_i h(x_i)\left(\frac{\rho_+ + \rho_-}{\rho_+}\right)\right) + \sum_{i \in I_-} \max\left(0, 1 - y_i h(x_i)\left(\frac{\rho_+ + \rho_-}{\rho_-}\right)\right)\right] \\ &+ \frac{4r}{m}\sqrt{\frac{m_+}{\rho_+^2} + \frac{m_-}{\rho_-^2}} + \sqrt{\frac{\log \log_2 \frac{2r_+}{\rho_+}}{m}} + \sqrt{\frac{\log \log_2 \frac{2r_-}{\rho_-}}{m}} + \sqrt{\frac{\log \frac{4}{\delta}}{2m}}.\end{aligned}$$

Now, since only the first term of the right-hand side depends on $w$, the bound suggests selecting $w$ as the solution of the following optimization problem:

$$\min_{\|w\|^2 \le \left(\frac{1}{\rho_+ + \rho_-}\right)^2} \frac{1}{m}\left[\sum_{i \in I_+} \max\left(0, 1 - y_i h(x_i)\left(\frac{\rho_+ + \rho_-}{\rho_+}\right)\right) + \sum_{i \in I_-} \max\left(0, 1 - y_i h(x_i)\left(\frac{\rho_+ + \rho_-}{\rho_-}\right)\right)\right].$$

Introducing a Lagrange variable $\lambda \ge 0$ and a free variable $\alpha = \frac{\rho_+}{\rho_+ + \rho_-} > 0$, the optimization problem can be written equivalently as

$$\min_w \lambda\|w\|^2 + \frac{1}{m}\left[\sum_{i \in I_+} \max\left(0, 1 - y_i \frac{w \cdot x_i}{\alpha}\right) + \sum_{i \in I_-} \max\left(0, 1 - y_i \frac{w \cdot x_i}{1 - \alpha}\right)\right], \tag{10}$$

where $\lambda$ and $\alpha$ can be selected via cross-validation. The resulting algorithm can be viewed as an extension of SVMs.

Note that while $\alpha$ can be freely searched over different values, we can search near the optimal values found in the separable case in (2). Also, the solution can actually be obtained using regular SVM by incorporating the $\alpha$ multipliers into the feature

vectors. Furthermore, we can replace the hinge loss with a general margin-based loss function $\Psi\colon u \mapsto \mathbb{R}_+$, and we can add a bias term $b > 0$ for the linear models if the data is not normalized:

$$\min_{w,b} \lambda \|w\|^2 + \frac{1}{m}\left[\sum_{i\in I_+} \Psi\left(y_i \frac{w \cdot x_i + b}{\alpha}\right) + \sum_{i\in I_-} \Psi\left(y_i \frac{w \cdot x_i + b}{1-\alpha}\right)\right], \tag{11}$$

For example, $\Psi$ can be chosen as the logistic loss function $u \mapsto \log_2(1 + e^{-u})$ or the exponential loss function $u \mapsto e^{-u}$.

### D.6. Proof of Theorem 4.1

**Theorem 4.1.** *Let $S \subseteq \{x\colon \|x\| \le r\}$ be a sample of size $m$ and let $\mathcal{H} = \{x \mapsto w \cdot x\colon \|w\| \le \Lambda\}$. Let $r_+ = \sup_{i\in I_+}\|x_i\|$ and $r_- = \sup_{i\in I_-}\|x_i\|$. Then, the following bound holds for all $h \in \mathcal{H}$:*

$$\widehat{\mathfrak{R}}_S^{\rho_+,\rho_-}(\mathcal{H}) \le \frac{\Lambda}{m}\sqrt{\frac{m_+ r_+^2}{\rho_+^2} + \frac{m_- r_-^2}{\rho_-^2}} \le \frac{\Lambda r}{m}\sqrt{\frac{m_+}{\rho_+^2} + \frac{m_-}{\rho_-^2}}.$$

*Proof.* The proof follows through a series of inequalities:

$$\widehat{\mathfrak{R}}_S^{\rho_+,\rho_-}(\mathcal{H})$$

$$= \frac{1}{m}\mathop{\mathbb{E}}_{\sigma}\left[\sup_{\|w\|\le\Lambda} w \cdot \left(\frac{1}{\rho_+}\left(\sum_{i\in I_+}\sigma_i x_i\right) + \frac{1}{\rho_-}\left(\sum_{i\in I_-}-\sigma_i x_i\right)\right)\right]$$

$$\le \frac{\Lambda}{m}\mathop{\mathbb{E}}_{\sigma}\left[\left\|\frac{1}{\rho_+}\left(\sum_{i\in I_+}\sigma_i x_i\right) + \frac{1}{\rho_-}\left(\sum_{i\in I_-}-\sigma_i x_i\right)\right\|\right] \le \frac{\Lambda}{m}\left[\mathop{\mathbb{E}}_{\sigma}\left[\left\|\frac{1}{\rho_+}\left(\sum_{i\in I_+}\sigma_i x_i\right) + \frac{1}{\rho_-}\left(\sum_{i\in I_-}-\sigma_i x_i\right)\right\|^2\right]\right]^{\frac{1}{2}}$$

$$\le \frac{\Lambda}{m}\left[\frac{1}{\rho_+^2}\sum_{i\in I_+}\|x_i\|^2 + \frac{1}{\rho_-^2}\sum_{i\in I_-}\|x_i\|^2\right]^{\frac{1}{2}} \le \frac{\Lambda}{m}\sqrt{\frac{m_+ r_+^2}{\rho_+^2} + \frac{m_- r_-^2}{\rho_-^2}} \le \frac{\Lambda r}{m}\sqrt{\frac{m_+}{\rho_+^2} + \frac{m_-}{\rho_-^2}}.$$

The first inequality makes use of the Cauchy-Schwarz inequality and the bound on $\|w\|$, the second follows by Jensen's inequality, the third by $\mathbb{E}[\sigma_i\sigma_j] = \mathbb{E}[\sigma_i]\mathbb{E}[\sigma_j] = 0$ for $i \ne j$, the fourth by $\sup_{i\in I_+}\|x_i\| = r_+$ and $\sup_{i\in I_-}\|x_i\| = r_-$, and the last one by $\|x_i\| \le r$. $\qquad\square$

## E. Extension to Multi-Class Classification

In this section, we extend the previous analysis and algorithm to multi-class classification. We will adopt the same notation and definitions as previously described, with some slight adjustments. In particular, we denote the multi-class label space by $\mathcal{Y} = [c] := \{1, \ldots, c\}$ and a hypothesis set of functions mapping from $\mathcal{X} \times \mathcal{Y}$ to $\mathbb{R}$ by $\mathcal{H}$. For a hypothesis $h \in \mathcal{H}$, the label $h(x)$ assigned to $x \in \mathcal{X}$ is the one with the largest score, defined as $h(x) = \arg\max_{y\in\mathcal{Y}} h(x, y)$, using the highest index for tie-breaking. For a labeled example $(x, y) \in \mathcal{X} \times \mathcal{Y}$, the *margin* $\rho_h(x, y)$ of a hypothesis $h \in \mathcal{H}$ is given by $\rho_h(x, y) = h(x, y) - \max_{y'\ne y} h(x, y')$, which is the difference between the score assigned to $(x, y)$ and that of the next-highest scoring label. We define the multi-class zero-one loss function as $\ell_{0-1}^{\mathrm{multi}} := 1_{h(x)\ne y}$. This is the target loss of interest in multi-class classification.

### E.1. Multi-Class Imbalanced Margin Loss

We first extend the class-imbalanced margin loss function to the multi-class setting. To account for different confidence margins for instances with different labels, we define the *multi-class class-imbalanced margin loss function* as follows:

**Definition E.1** (Multi-class class-imbalanced margin loss)**.** For any $\boldsymbol{\rho} = [\rho_k]_{k\in[c]}$, the *multi-class class-imbalanced $\boldsymbol{\rho}$-margin loss* is the function $\mathsf{L}_{\boldsymbol{\rho}}\colon \mathcal{H}_{\mathrm{all}} \times \mathcal{X} \times \mathcal{Y} \to \mathbb{R}$, defined as follows:

$$\mathsf{L}_{\boldsymbol{\rho}}(h, x, y) = \sum_{k=1}^{c} \Phi_{\rho_k}(\rho_h(x, y))1_{y=k}. \tag{12}$$

The main margin bounds in this section are expressed in terms of this loss function. The parameters $\rho_k > 0$, for $k \in [c]$, represent the confidence margins imposed by a hypothesis $h$ for instances labeled $k$. The following result provides an equivalent expression for the class-imbalanced margin loss function. The proof is included in Appendix F.1.

**Lemma 5.2.** *The multi-class class-imbalanced $\boldsymbol{\rho}$-margin loss can be equivalently expressed as follows:*

$$\mathsf{L}_{\boldsymbol{\rho}}(h, x, y) = \sum_{k=1}^{c} \Phi_{\rho_k}(\rho_h(x, y)) 1_{\mathsf{h}(x)=k}.$$

## E.2. $\mathcal{H}$-Consistency

The following result provides a strong consistency guarantee for the multi-class class-imbalanced margin loss introduced in relation to the multi-class zero-one loss. We say a hypothesis set is complete when the scoring values spanned by $\mathcal{H}$ for each instance cover $\mathbb{R}$: for all $(x, y) \in \mathcal{X} \times \mathcal{Y}$, $\{h(x, y): h \in \mathcal{H}\} = \mathbb{R}$.

**Theorem 5.3** ($\mathcal{H}$-Consistency bound for multi-class class-imbalanced margin loss). *Let $\mathcal{H}$ be a complete hypothesis set. Then, for all $h \in \mathcal{H}$ and $\boldsymbol{\rho} = [\rho_k]_{k \in [c]} > \mathbf{0}$, the following bound holds:*

$$\mathcal{R}_{\ell_{0-1}^{\mathrm{multi}}}(h) - \mathcal{R}_{\ell_{0-1}^{\mathrm{multi}}}^*(\mathcal{H}) + \mathcal{M}_{\ell_{0-1}^{\mathrm{multi}}}(\mathcal{H}) \leq \mathcal{R}_{\mathsf{L}_{\boldsymbol{\rho}}}(h) - \mathcal{R}_{\mathsf{L}_{\boldsymbol{\rho}}}^*(\mathcal{H}) + \mathcal{M}_{\mathsf{L}_{\boldsymbol{\rho}}}(\mathcal{H}). \tag{4}$$

The proof is included in Appendix F.2. The next section presents generalization bounds based on the empirical multi-class class-imbalanced margin loss, along with the $\boldsymbol{\rho}$-*class-sensitive Rademacher complexity* and its empirical counterpart defined below. Given a sample $S = (x_1, \ldots, x_m)$, for any $k \in [c]$, we define $I_k = \{i \in \{1, \ldots, m\} \mid y_i = k\}$ and $m_k = |I_k|$ as the number of instances labeled $k$.

**Definition E.2** ($\boldsymbol{\rho}$-class-sensitive Rademacher complexity). Let $\mathcal{H}$ be a family of functions mapping from $\mathcal{X} \times \mathcal{Y}$ to $\mathbb{R}$ and $S = ((x_1, y_1) \ldots, (x_m, y_m))$ a fixed sample of size $m$ with elements in $\mathcal{X} \times \mathcal{Y}$. Fix $\boldsymbol{\rho} = [\rho_k]_{k \in [c]} > \mathbf{0}$. Then, the *empirical $\boldsymbol{\rho}$-class-sensitive Rademacher complexity of* $\mathcal{H}$ with respect to the sample $S$ is defined as:

$$\widehat{\mathfrak{R}}_S^{\boldsymbol{\rho}}(\mathcal{H}) = \frac{1}{m} \mathop{\mathbb{E}}_{\epsilon}\left[\sup_{h \in \mathcal{H}}\left\{\sum_{k=1}^{c} \sum_{i \in I_k} \sum_{y \in \mathcal{Y}} \epsilon_{iy} \frac{h(x_i, y)}{\rho_k}\right\}\right], \tag{13}$$

where $\epsilon = (\epsilon_{iy})_{i,y}$ with $\epsilon_{iy}$s being independent variables uniformly distributed over $\{-1, +1\}$. For any integer $m \geq 1$, the $\boldsymbol{\rho}$-*class-sensitive Rademacher complexity of* $\mathcal{H}$ is the expectation of the empirical $\boldsymbol{\rho}$-class-sensitive Rademacher complexity over all samples of size $m$ drawn according to $\mathcal{D}$: $\mathfrak{R}_m^{\boldsymbol{\rho}}(\mathcal{H}) = \mathbb{E}_{S \sim \mathcal{D}^m}[\widehat{\mathfrak{R}}_S^{\boldsymbol{\rho}}(\mathcal{H})]$.

## E.3. Margin-Based Guarantees

Next, we will prove a general margin-based generalization bound, which will serve as the foundation for deriving new algorithms for imbalanced multi-class classification.

Given a sample $S = (x_1, \ldots, x_m)$ and a hypothesis $h$, the *empirical multi-class class-imbalanced margin loss* is defined by $\widehat{\mathcal{R}}_S^{\boldsymbol{\rho}}(h) = \frac{1}{m} \sum_{i=1}^{m} \mathsf{L}_{\boldsymbol{\rho}}(h, x_i, y_i)$. Note that the multi-class zero-one loss function $\ell_{0-1}^{\mathrm{multi}}$ is upper bounded by the multi-class class-imbalanced margin loss $\mathsf{L}_{\boldsymbol{\rho}}$: $\mathcal{R}_{\ell_{0-1}^{\mathrm{multi}}}(h) \leq \mathcal{R}_{\mathsf{L}_{\boldsymbol{\rho}}}(h)$.

**Theorem E.3** (Margin bound for imbalanced multi-class classification). *Let $\mathcal{H}$ be a set of real-valued functions. Fix $\rho_k > 0$ for $k \in [c]$, then, for any $\delta > 0$, with probability at least $1 - \delta$, each of the following holds for all $h \in \mathcal{H}$:*

$$\mathcal{R}_{\ell_{0-1}^{\mathrm{multi}}}(h) \leq \widehat{\mathcal{R}}_S^{\boldsymbol{\rho}}(h) + 4\sqrt{2c}\,\mathfrak{R}_m^{\boldsymbol{\rho}}(\mathcal{H}) + \sqrt{\frac{\log \frac{1}{\delta}}{2m}}$$

$$\mathcal{R}_{\ell_{0-1}^{\mathrm{multi}}}(h) \leq \widehat{\mathcal{R}}_S^{\boldsymbol{\rho}}(h) + 4\sqrt{2c}\,\widehat{\mathfrak{R}}_S^{\boldsymbol{\rho}}(\mathcal{H}) + 3\sqrt{\frac{\log \frac{2}{\delta}}{2m}}.$$

The proof is presented in Appendix F.3. As in Theorem D.1, these bounds can be generalized to hold uniformly for all $\rho_k \in (0, 1]$, at the cost of additional terms $\sqrt{\frac{\log \log_2 \frac{2}{\rho_k}}{m}}$ for $k \in [c]$, as shown in Theorem F.2 in Appendix F.5.

As for margin bounds in imbalanced binary classification, they show the conflict between two terms: the larger the desired margins $\boldsymbol{\rho}$, the smaller the second term, at the price of a larger empirical multi-class class-imbalanced margin loss $\widehat{\mathcal{R}}_S^{\boldsymbol{\rho}}$. Note, however, that here there is additionally a dependency on the number of classes $c$. This suggests either weak guarantees when learning with a large number of classes or the need for even larger margins $\boldsymbol{\rho}$ for which the empirical multi-class class-imbalanced margin loss would be small.

### E.4. General Multi-Class Classification Algorithms

Here, we derive IMMAX algorithms for multi-class classification in imbalanced settings, building on the theoretical analysis from the previous section.

Let $\Phi$ be a feature mapping from $\mathcal{X} \times \mathcal{Y}$ to $\mathbb{R}^d$. Let $S \subseteq \{(x,y) \colon \|\Phi(x,y)\| \leq r\}$ denote a sample of size $m$, for some appropriate norm $\|\cdot\|$ on $\mathbb{R}^d$. Define $r_k = \sup_{i \in I_k, y \in \mathcal{Y}} \|\Phi(x_i, y)\|$, for any $k \in [c]$. As in the binary case, we assume that the empirical class-sensitive Rademacher complexity $\widehat{\mathfrak{R}}_S^\rho(\mathcal{H})$ can be bounded as:

$$\widehat{\mathfrak{R}}_S^\rho(\mathcal{H}) \leq \frac{\Lambda_{\mathcal{H}} \sqrt{c}}{m} \sqrt{\sum_{k=1}^c \frac{m_k r_k^2}{\rho_k^2}} \leq \frac{\Lambda_{\mathcal{H}} r \sqrt{c}}{m} \sqrt{\sum_{k=1}^c \frac{m_k}{\rho_k^2}},$$

where $\Lambda_{\mathcal{H}}$ depends on the complexity of the hypothesis set $\mathcal{H}$. This bound holds for many commonly used hypothesis sets. For a family of neural networks, $\Lambda_{\mathcal{H}}$ can be expressed as a Frobenius norm (Cortes et al., 2017; Neyshabur et al., 2015) or spectral norm complexity with respect to reference weight matrices (Bartlett et al., 2017). Additionally, Theorems F.3 and F.4 in Appendix F.6 address kernel-based hypotheses. More generally, for the analysis that follows, we will assume that $\mathcal{H}$ can be defined by $\mathcal{H} = \{h \in \overline{\mathcal{H}} \colon \|h\| \leq \Lambda_{\mathcal{H}}\}$, for some appropriate norm $\|\cdot\|$ on some space $\overline{\mathcal{H}}$. Combining such an upper bound and Theorem E.3 or Theorem F.2, gives directly the following general margin bound:

$$\mathcal{R}_{\ell_{0-1}^{\mathrm{multi}}}(h) \leq \widehat{\mathfrak{R}}_S^\rho(h) + \frac{4\sqrt{2}\Lambda_{\mathcal{H}} rc}{m} \sqrt{\sum_{k=1}^c \frac{m_k}{\rho_k^2}} + O\left(\frac{1}{\sqrt{m}}\right),$$

where the last term includes the log-log terms and the $\delta$-confidence term. Let $\Psi$ be a non-increasing convex function such that $\Phi_\rho(u) \leq \Psi\left(\frac{u}{\rho}\right)$ for all $u \in \mathbb{R}$. Then, since $\Phi_\rho$ is non-increasing, for any $(x,k)$, we have: $\Phi_\rho(\rho_h(x,k)) = \max_{j \neq k} \Phi_\rho(h(x,k) - h(x,j))$. This suggests a regularization-based algorithm of the following form:

$$\min_{h \in \overline{\mathcal{H}}} \lambda \|h\|^2 + \frac{1}{m}\left[\sum_{k=1}^c \sum_{i \in I_k} \max_{j \neq k} \Psi\left(\frac{h(x,k) - h(x,j)}{\rho_k}\right)\right], \tag{14}$$

where $\lambda$ and $\rho_k$s are chosen via cross-validation. In particular, choosing $\Psi$ to be the logistic loss and upper-bounding the maximum by a sum yields the following form for our IMMAX (*Imbalanced Margin Maximization*) algorithm:

$$\min_{h \in \overline{\mathcal{H}}} \lambda \|h\|^2 + \frac{1}{m} \sum_{k=1}^c \sum_{i \in I_k} \log\left[\sum_{j=1}^c \exp\left(\frac{h(x_i,j) - h(x_i,k)}{\rho_k}\right)\right], \tag{15}$$

where $\lambda$ and $\rho_k$s are chosen via cross-validation. Let $\rho = \sum_{k=1}^c \rho_k$ and $\overline{r} = \left[\sum_{k=1}^c m_k^{\frac{1}{3}} r_{k,2}^{\frac{2}{3}}\right]^{\frac{3}{2}}$. Using Lemma F.1 (Appendix F.4), the expression under the square root in the second term of the generalization bound can be reformulated in terms of the Rényi divergence of order 3 as: $\sum_{k=1}^c \frac{m_k r_{k,2}^2}{\rho_k^2} = \frac{\overline{r}^2}{\rho^2} e^{2D_3(\mathsf{r} \| \frac{\rho}{\rho})}$, where $\mathsf{r} = \left[\frac{m_k^{\frac{1}{3}} r_{k,2}^{\frac{2}{3}}}{\overline{r}^{\frac{2}{3}}}\right]_k$. Thus, while $\rho_k$s can be freely searched over a range of values in our general algorithm, it may be beneficial to focus the search for the vector $[\rho_k/\rho]_k$ near $\mathsf{r}$. This strictly generalizes our binary classification results and the analysis of the separable case.

When the number of classes $c$ is very large, the search space can be further reduced by constraining the $\rho_k$ values for underrepresented classes to be identical and allowing distinct $\rho_k$ values only for the most frequently occurring classes.

## F. Multi-Class Classification: Proofs

### F.1. Proof of Lemma 5.2

**Lemma 5.2.** *The multi-class class-imbalanced $\rho$-margin loss can be equivalently expressed as follows:*

$$\mathsf{L}_{\boldsymbol{\rho}}(h,x,y) = \sum_{k=1}^c \Phi_{\rho_k}(\rho_h(x,y)) 1_{\mathsf{h}(x)=k}.$$

*Proof.* When $\rho_h(x,y) \leq 0$, we have $\Phi_{\rho_k}(\rho_h(x,y)) = 1$ for any $k \in [c]$, so the equality holds. When $\rho_h(x,y) > 0$, we have $y = k \iff \rho_h(x,k) > 0 \iff \mathsf{h}(x) = k$, which also implies the equality. $\qquad\square$

## F.2. Proof of Theorem 5.3

**Theorem 5.3** ($\mathcal{H}$-Consistency bound for multi-class class-imbalanced margin loss)**.** *Let $\mathcal{H}$ be a complete hypothesis set. Then, for all $h \in \mathcal{H}$ and $\boldsymbol{\rho} = [\rho_k]_{k \in [c]} > \mathbf{0}$, the following bound holds:*

$$\mathcal{R}_{\ell_{0-1}^{\mathrm{multi}}}(h) - \mathcal{R}^*_{\ell_{0-1}^{\mathrm{multi}}}(\mathcal{H}) + \mathcal{M}_{\ell_{0-1}^{\mathrm{multi}}}(\mathcal{H}) \leq \mathcal{R}_{\mathsf{L}_{\boldsymbol{\rho}}}(h) - \mathcal{R}^*_{\mathsf{L}_{\boldsymbol{\rho}}}(\mathcal{H}) + \mathcal{M}_{\mathsf{L}_{\boldsymbol{\rho}}}(\mathcal{H}). \tag{4}$$

*Proof.* Let $p(y|x) = \mathbb{P}(Y = y \,|\, X = x)$ denote the conditional probability that $Y = y$ given $X = x$. Then, the conditional error and the best-in-class conditional error of the zero-one loss can be expressed as follows:

$$\mathbb{E}_y\big[\ell_{0-1}^{\mathrm{multi}}(h, x, y) \mid x\big] = \sum_{y \in \mathcal{Y}} p(y|x) 1_{\mathsf{h}(x) \neq y} = 1 - p(\mathsf{h}(x)|x),$$

$$\inf_{h \in \mathcal{H}} \mathbb{E}_y\big[\ell_{0-1}^{\mathrm{multi}}(h, x, y) \mid x\big] = 1 - \max_{y \in \mathcal{Y}} p(y|x).$$

Furthermore, the difference between the two terms is given by:

$$\mathbb{E}_y\big[\ell_{0-1}^{\mathrm{multi}}(h, x, y) \mid x\big] - \inf_{h \in \mathcal{H}} \mathbb{E}_y\big[\ell_{0-1}^{\mathrm{multi}}(h, x, y) \mid x\big] = \max_{y \in \mathcal{Y}} p(y|x) - p(\mathsf{h}(x)|x).$$

For the multi-class class-imbalanced margin loss, the conditional error can be expressed as follows:

$$\begin{aligned}
\mathbb{E}_y[\mathsf{L}_{\boldsymbol{\rho}}(h, x, y) \mid x] &= \sum_{y \in \mathcal{Y}} p(y|x) \Phi_{\rho_y}(\rho_h(x, y)) \\
&= \sum_{y \in \mathcal{Y}} p(y|x) \min\left(1, \max\left(0, 1 - \frac{\rho_h(x, y)}{\rho_y}\right)\right) \\
&= 1 - p(\mathsf{h}(x)|x) + p(\mathsf{h}(x)|x) \max\left(0, 1 - \frac{\rho_h(x, \mathsf{h}(x))}{\rho_{\mathsf{h}(x)}}\right) \\
&= 1 - p(\mathsf{h}(x)|x) \min\left(1, \frac{\rho_h(x, \mathsf{h}(x))}{\rho_{\mathsf{h}(x)}}\right).
\end{aligned}$$

Thus, the best-in-class conditional error can be expressed as follows:

$$\inf_{h \in \mathcal{H}} \mathbb{E}_y[\mathsf{L}_{\boldsymbol{\rho}}(h, x, y) \mid x] = 1 - \max_{y \in \mathcal{Y}} p(y|x).$$

The difference between the two terms is given by:

$$\begin{aligned}
\mathbb{E}_y[\mathsf{L}_{\boldsymbol{\rho}}(h, x, y) \mid x] - \inf_{h \in \mathcal{H}} \mathbb{E}_y[\mathsf{L}_{\boldsymbol{\rho}}(h, x, y) \mid x] &= \max_{y \in \mathcal{Y}} p(y|x) - p(\mathsf{h}(x)|x) \min\left(1, \frac{\rho_h(x, \mathsf{h}(x))}{\rho_{\mathsf{h}(x)}}\right) \\
&\geq \max_{y \in \mathcal{Y}} p(y|x) - p(\mathsf{h}(x)|x) \\
&= \mathbb{E}_y\big[\ell_{0-1}^{\mathrm{multi}}(h, x, y) \mid x\big] - \inf_{h \in \mathcal{H}} \mathbb{E}_y\big[\ell_{0-1}^{\mathrm{multi}}(h, x, y) \mid x\big].
\end{aligned}$$

By taking the expectation of both sides, we obtain:

$$\mathcal{R}_{\ell_{0-1}^{\mathrm{multi}}}(h) - \mathcal{R}^*_{\ell_{0-1}^{\mathrm{multi}}}(\mathcal{H}) + \mathcal{M}_{\ell_{0-1}^{\mathrm{multi}}}(\mathcal{H}) \leq \mathcal{R}_{\mathsf{L}_{\boldsymbol{\rho}}}(h) - \mathcal{R}^*_{\mathsf{L}_{\boldsymbol{\rho}}}(\mathcal{H}) + \mathcal{M}_{\mathsf{L}_{\boldsymbol{\rho}}}(\mathcal{H}),$$

which completes the proof. $\qquad\square$

## F.3. Proof of Theorem E.3

*Proof.* Consider the family of functions taking values in $[0, 1]$:

$$\widetilde{\mathcal{H}} = \{z = (x, y) \mapsto \mathsf{L}_{\boldsymbol{\rho}}(h, x, y) : h \in \mathcal{H}\}.$$

By (Mohri et al., 2018, Theorem 3.3), with probability at least $1 - \delta$, for all $g \in \widetilde{\mathcal{H}}$,

$$\mathbb{E}[g(z)] \leq \frac{1}{m} \sum_{i=1}^m g(z_i) + 2\widehat{\mathfrak{R}}_S(\widetilde{\mathcal{H}}) + 3\sqrt{\frac{\log \frac{2}{\delta}}{2m}},$$

and thus, for all $h \in \mathcal{H}$,

$$\mathbb{E}[\mathsf{L}_{\boldsymbol{\rho}}(h, x, y)] \leq \widehat{\mathcal{R}}_S^{\rho}(h) + 2\widehat{\mathfrak{R}}_S(\widetilde{\mathcal{H}}) + 3\sqrt{\frac{\log\frac{2}{\delta}}{2m}}.$$

Since $\mathcal{R}_{\ell_{0-1}^{\mathrm{multi}}}(h) \leq \mathcal{R}_{\mathsf{L}_{\boldsymbol{\rho}}}(h) = \mathbb{E}[\mathsf{L}_{\boldsymbol{\rho}}(h, x, y)]$, we have

$$\mathcal{R}_{\ell_{0-1}^{\mathrm{multi}}}(h) \leq \widehat{\mathcal{R}}_S^{\rho}(h) + 2\widehat{\mathfrak{R}}_S(\widetilde{\mathcal{H}}) + 3\sqrt{\frac{\log\frac{2}{\delta}}{2m}}.$$

For convenience, we define $\rho(i) = \sum_{k=1}^c \rho_k 1_{i \in I_k}$ for $i = 1, \ldots, m$. Since $\Phi_\rho$ is $\frac{1}{\rho}$-Lipschitz, by (Mohri et al., 2018, Lemma 5.7), $\widehat{\mathfrak{R}}_S(\widetilde{\mathcal{H}})$ can be rewritten as follows:

$$\begin{aligned}
\widehat{\mathfrak{R}}_S(\widetilde{\mathcal{H}}) &= \frac{1}{m}\,\mathbb{E}_\sigma\!\left[\sup_{h \in \mathcal{H}}\sum_{i=1}^m \sigma_i \mathsf{L}_{\boldsymbol{\rho}}(h, x_i, y_i)\right] \\
&= \frac{1}{m}\,\mathbb{E}_\sigma\!\left[\sup_{h \in \mathcal{H}}\sum_{i=1}^m \sigma_i\left[\sum_{k=1}^c \Phi_{\rho_k}(\rho_h(x_i, y_i))1_{y_i=k}\right]\right] \\
&\leq \frac{1}{m}\,\mathbb{E}_\sigma\!\left[\sup_{h \in \mathcal{H}}\left\{\sum_{i=1}^m \sigma_i\frac{\rho_h(x_i, y_i)}{\rho(i)}\right\}\right] \\
&= \frac{1}{m}\,\mathbb{E}_\sigma\!\left[\sup_{h \in \mathcal{H}}\left\{\sum_{i=1}^m \sigma_i\frac{h(x_i, y_i) - \max_{y' \neq y_i} h(x_i, y')}{\rho(i)}\right\}\right] \\
&\leq \frac{1}{m}\,\mathbb{E}_\sigma\!\left[\sup_{h \in \mathcal{H}}\left\{\sum_{i=1}^m \sigma_i\frac{h(x_i, y_i)}{\rho(i)}\right\}\right] + \frac{1}{m}\,\mathbb{E}_\sigma\!\left[\sup_{h \in \mathcal{H}}\left\{\sum_{i=1}^m \sigma_i\frac{\max_{y' \neq y_i} h(x_i, y')}{\rho(i)}\right\}\right].
\end{aligned}$$

Now we bound the second term above. For any $i = 1, \ldots, m$, consider the mapping $\Psi_i \colon h \mapsto \frac{\max_{y' \neq y_i} h(x_i, y')}{\rho(i)}$. Then, for any $h, h' \in \mathcal{H}$, we have

$$\begin{aligned}
|\Psi_i(h) - \Psi_i(h')| &\leq \max_{y' \neq y_i}\frac{|h(x_i, y') - h'(x_i, y')|}{\rho(i)} \\
&\leq \frac{1}{\rho(i)}\sum_{y \in \mathcal{Y}}|h(x_i, y) - h'(x_i, y)| \\
&\leq \frac{\sqrt{c}}{\rho(i)}\sqrt{\sum_{y \in \mathcal{Y}}|h(x_i, y) - h'(x_i, y)|^2}.
\end{aligned}$$

Thus, $\Psi_i$ is $\frac{\sqrt{c}}{\rho(i)}$-Lipschitz with respect to the $\|\cdot\|_2$ norm. Thus, by (Cortes et al., 2016, Lemma 5),

$$\begin{aligned}
\frac{1}{m}\,\mathbb{E}_\sigma\!\left[\sup_{h \in \mathcal{H}}\left\{\sum_{i=1}^m \sigma_i\frac{\max_{y' \neq y_i} h(x_i, y')}{\rho(i)}\right\}\right] &\leq \frac{\sqrt{2}}{m}\,\mathbb{E}_\sigma\!\left[\sup_{h \in \mathcal{H}}\left\{\sum_{i=1}^m\sum_{y \in \mathcal{Y}} \sigma_{iy}\frac{\sqrt{c}}{\rho(i)}h(x_i, y)\right\}\right] \\
&= \frac{\sqrt{2c}}{m}\,\mathbb{E}_\epsilon\!\left[\sup_{h \in \mathcal{H}}\left\{\sum_{k=1}^c\sum_{i \in I_k}\sum_{y \in \mathcal{Y}} \epsilon_{iy}\frac{h(x_i, y)}{\rho_k}\right\}\right] \\
&= \sqrt{2c}\,\widehat{\mathfrak{R}}_S^{\rho}(\mathcal{H}).
\end{aligned}$$

We can proceed similarly with the first term to obtain

$$\frac{1}{m}\,\mathbb{E}_\sigma\!\left[\sup_{h \in \mathcal{H}}\left\{\sum_{i=1}^m \sigma_i\frac{h(x_i, y_i)}{\rho(i)}\right\}\right] \leq \sqrt{2c}\,\widehat{\mathfrak{R}}_S^{\rho}(\mathcal{H}).$$

Thus, $\widehat{\mathfrak{R}}_S(\widetilde{\mathcal{H}})$ can be upper bounded as follows:

$$\widehat{\mathfrak{R}}_S(\widetilde{\mathcal{H}}) \leq 2\sqrt{2c}\,\widehat{\mathfrak{R}}_S^{\rho}(\mathcal{H}).$$

This proves the second inequality. The first inequality, can be derived in the same way by using the first inequality of (Mohri et al., 2018, Theorem 3.3). $\qquad\square$

## F.4. Analysis of the Second Term in the Generalization Bound

In this section, we analyze the second term of the generalization bound in terms of the Rényi entropy of order 3.

Recall that the Rényi divergence of positive order $\alpha$ between two distributions $\mathsf{p}$ and $\mathsf{q}$ with support $[c]$ is defined as:

$$\mathsf{D}_\alpha(\mathsf{p} \,\|\, \mathsf{q}) = \frac{1}{\alpha - 1} \log \left[ \sum_{k=1}^{c} \mathsf{p}_k^\alpha \mathsf{q}_k^{1-\alpha} \right],$$

with the conventions $\frac{0}{0} = 0$ and $\frac{x}{0} = \infty$ for $x > 0$. This definition extends to $\alpha \in \{0, 1, \infty\}$ by taking appropriate limits. In particular, $\mathsf{D}_1$ corresponds to the relative entropy (KL divergence).

**Lemma F.1.** *Let $\rho = \sum_{k=1}^{c} \rho_k$ and $\overline{r} = \left[ \sum_{k=1}^{c} m_k^{\frac{1}{3}} r_{k,2}^{\frac{2}{3}} \right]^{\frac{3}{2}}$. Then, the following identity holds:*

$$\sum_{k=1}^{c} \frac{m_k r_{k,2}^2}{\rho_k^2} = \frac{\overline{r}^2}{\rho^2} e^{2\mathsf{D}_3\left(\mathsf{r} \,\|\, \frac{\rho}{\rho}\right)},$$

*where $\mathsf{r} = \left[ \dfrac{m_k^{\frac{1}{3}} r_{k,2}^{\frac{2}{3}}}{\overline{r}^{\frac{2}{3}}} \right]_{k \in [c]}$.*

*Proof.* The expression can be rewritten as follows after putting $\frac{\overline{r}^2}{\rho^2} \sum_{k=1}^{c}$ in factor:

$$
\begin{aligned}
\sum_{k=1}^{c} \frac{m_k r_{k,2}^2}{\rho_k^2} &= \frac{\overline{r}^2}{\rho^2} \sum_{k=1}^{c} \frac{\left( \frac{\sqrt{m_k} r_{k,2}}{\overline{r}} \right)^2}{\left( \frac{\rho_k}{\rho} \right)^2} \\
&= \frac{\overline{r}^2}{\rho^2} \sum_{k=1}^{c} \frac{\left( \frac{m_k^{\frac{1}{3}} r_{k,2}^{\frac{2}{3}}}{\overline{r}^{\frac{2}{3}}} \right)^3}{\left( \frac{\rho_k}{\rho} \right)^{3-1}} \\
&= \frac{\overline{r}^2}{\rho^2} \exp\left\{ 2\, \mathsf{D}_3\left( \left[ \frac{m_k^{\frac{1}{3}} r_{k,2}^{\frac{2}{3}}}{\overline{r}^{\frac{2}{3}}} \right]_{k \in [c]} \,\middle\|\, \left[ \frac{\rho_k}{\rho} \right]_{k \in [c]} \right) \right\}.
\end{aligned}
$$

This completes the proof. $\qquad\square$

The lemma suggests that for fixed $\rho$, choosing $[\rho_k/\rho]_k$ close to $\mathsf{r}$ tends to minimize the second term of the generalization bound. Specifically, in the separable case where the empirical margin loss is zero, this analysis provides guidance on selecting $\rho_k$s. The optimal values in this scenario align with those derived in the analysis of the separable binary case.

## F.5. Uniform Margin Bound for Imbalanced Multi-Class Classification

**Theorem F.2** (Uniform margin bound for imbalanced multi-class classification)**.** *Let $\mathcal{H}$ be a set of real-valued functions. Fix $r_k > 0$ for $k \in [c]$. Then, for any $\delta > 0$, with probability at least $1 - \delta$, each of the following holds for all $h \in \mathcal{H}$ and $\rho_k \in (0, r_k]$ with $k \in [c]$:*

$$\mathcal{R}_{\ell_{0-1}^{\mathrm{multi}}}(h) \leq \widehat{\mathcal{R}}_S^\rho(h) + 4c\sqrt{2c}\,\mathfrak{R}_m^\rho(\mathcal{H}) + \sum_{k=1}^{c} \sqrt{\frac{\log \log_2 \frac{2r_k}{\rho_k}}{m}} + \sqrt{\frac{\log \frac{2c}{\delta}}{2m}}$$

$$\mathcal{R}_{\ell_{0-1}^{\mathrm{multi}}}(h) \leq \widehat{\mathcal{R}}_S^\rho(h) + 4c\sqrt{2c}\,\widehat{\mathfrak{R}}_S^\rho(\mathcal{H}) + \sum_{k=1}^{c} \sqrt{\frac{\log \log_2 \frac{2r_k}{\rho_k}}{m}} + 3\sqrt{\frac{\log \frac{2^{c+1}}{\delta}}{2m}}.$$

## F.6. Kernel-Based Hypotheses

For some hypothesis sets, a simpler upper bound can be derived for the $\rho$-class-sensitive Rademacher complexity of $\mathcal{H}$, thereby making Theorems E.3 and F.2 more explicit. We will show this for kernel-based hypotheses. Let $K: \mathcal{X} \times \mathcal{X} \to \mathbb{R}$ be a PDS kernel and let $\Phi: \mathcal{X} \to \mathbb{H}$ be a feature mapping associated to $K$. We consider kernel-based hypotheses with bounded weight vector: $\mathcal{H}_p = \left\{ (x, y) \mapsto w \cdot \Phi(x, y): w \in \mathbb{R}^d, \|w\|_p \leq \Lambda_p \right\}$, where $\Phi(x, y) = (\Phi_1(x, y), \ldots, \Phi_d(x, y))^\top$ is a $d$-dimensional feature vector. A similar analysis can be extended to hypotheses of the form $(x, y) \mapsto w_y \cdot \Phi(x, y)$, where $\|w_y\|_p \leq \Lambda_p$, based on $c$ weight vectors $w_1, \ldots, w_c \in \mathbb{R}^d$. The empirical $\rho$-class-sensitive Rademacher complexity of $\mathcal{H}_p$ with $p = 1$ and $p = 2$ can be bounded as follows.

**Theorem F.3.** *Consider* $\mathcal{H}_1 = \left\{ (x, y) \mapsto w \cdot \Phi(x, y): w \in \mathbb{R}^d, \|w\|_1 \leq \Lambda_1 \right\}$. *Let* $r_{k,\infty} = \sup_{i \in I_k, y \in \mathcal{Y}} \|\Phi(x_i, y)\|_\infty$, *for any* $k \in [c]$. *Then, the following bound holds for all* $h \in \mathcal{H}$*:*

$$\widehat{\mathfrak{R}}_S^\rho(\mathcal{H}_1) \leq \frac{\Lambda_1 \sqrt{2c}}{m} \sqrt{\sum_{k=1}^c \frac{m_k r_{k,\infty}^2}{\rho_k^2} \log(2d)}.$$

**Theorem F.4.** *Consider* $\mathcal{H}_2 = \left\{ (x, y) \mapsto w \cdot \Phi(x, y): w \in \mathbb{R}^d, \|w\|_2 \leq \Lambda_2 \right\}$. *Let* $r_{k,2} = \sup_{i \in I_k, y \in \mathcal{Y}} \|\Phi(x_i, y)\|_2$, *for any* $k \in [c]$. *Then, the following bound holds for all* $h \in \mathcal{H}$*:*

$$\widehat{\mathfrak{R}}_S^\rho(\mathcal{H}_2) \leq \frac{\Lambda_2 \sqrt{c}}{m} \sqrt{\sum_{k=1}^c \frac{m_k r_{k,2}^2}{\rho_k^2}}.$$

The proofs of Theorems F.3 and F.4 are included in Appendix F.7. Combining Theorem F.3 or Theorem F.4 with Theorem E.3 directly gives the following general margin bounds for kernel-based hypotheses with bounded weighted vectors, respectively.

**Corollary F.5.** *Consider* $\mathcal{H}_1 = \left\{ (x, y) \mapsto w \cdot \Phi(x, y): w \in \mathbb{R}^d, \|w\|_1 \leq \Lambda_1 \right\}$. *Let* $r_{k,\infty} = \sup_{i \in I_k, y \in \mathcal{Y}} \|\Phi(x_i, y)\|_\infty$, *for any* $k \in [c]$. *Fix* $\rho_k > 0$ *for* $k \in [c]$*, then, for any* $\delta > 0$*, with probability at least* $1 - \delta$ *over the choice of a sample* $S$ *of size* $m$*, the following holds for any* $h \in \mathcal{H}$*:*

$$\mathcal{R}_{\ell_{0-1}^{\mathrm{multi}}}(h) \leq \widehat{\mathfrak{R}}_S^\rho(h) + \frac{8\Lambda_1 c}{m} \sqrt{\sum_{k=1}^c \frac{m_k r_{k,\infty}^2}{\rho_k^2} \log(2d)} + \sqrt{\frac{\log \frac{1}{\delta}}{2m}}.$$

**Corollary F.6.** *Consider* $\mathcal{H}_2 = \left\{ (x, y) \mapsto w \cdot \Phi(x, y): w \in \mathbb{R}^d, \|w\|_2 \leq \Lambda_2 \right\}$. *Let* $r_{k,2} = \sup_{i \in I_k, y \in \mathcal{Y}} \|\Phi(x_i, y)\|_2$, *for any* $k \in [c]$. *Fix* $\rho_k > 0$ *for* $k \in [c]$*, then, for any* $\delta > 0$*, with probability at least* $1 - \delta$ *over the choice of a sample* $S$ *of size* $m$*, the following holds for any* $h \in \mathcal{H}$*:*

$$\mathcal{R}_{\ell_{0-1}^{\mathrm{multi}}}(h) \leq \widehat{\mathfrak{R}}_S^\rho(h) + \frac{4\sqrt{2}\Lambda_2 c}{m} \sqrt{\sum_{k=1}^c \frac{m_k r_{k,2}^2}{\rho_k^2}} + \sqrt{\frac{\log \frac{1}{\delta}}{2m}}.$$

As with Theorem E.3, the bounds of these corollaries can be generalized to hold uniformly for all $\rho_k \in (0, 1]$ with $k \in [c]$, at the cost of additional terms $\sqrt{\frac{\log \log_2 \frac{2}{\rho_k}}{m}}$ for $k \in [c]$ by combining Theorem F.3 or Theorem F.4 with Theorem F.2, respectively. Next, we describe an algorithm that can be derived directly from the theoretical guarantees presented above.

The guarantee of Corollary F.6 and it generalization to a uniform bound can be expressed as: for any $\delta > 0$, with probability at least $1 - \delta$, for all $h \in \mathcal{H}_2 = \left\{ (x, y) \mapsto w \cdot \Phi(x, y): w \in \mathbb{R}^d, \|w\|_2 \leq \Lambda_2 \right\}$,

$$\mathcal{R}_{\ell_{0-1}^{\mathrm{multi}}}(h) \leq \frac{1}{m} \left[ \sum_{k=1}^c \sum_{i \in I_k} \max\left(0, 1 - \frac{\rho_w(x_i, k)}{\rho_k}\right) \right] + \frac{4\sqrt{2}\Lambda_2 c}{m} \sqrt{\sum_{k=1}^c \frac{m_k r_{k,2}^2}{\rho_k^2}} + O\left(\frac{1}{\sqrt{m}}\right).$$

where $\rho_w(x, k) = w \cdot \Phi(x_i, k) - \max_{y' \neq k}(w \cdot \Phi(x_i, y'))$, and we used the fact that the $\rho$-margin loss function is upper bounded by the $\rho$-hinge loss.

This suggests a regularization-based algorithm of the following form:

$$\min_{w \in \mathbb{R}^d} \lambda \|w\|^2 + \frac{1}{m} \left[ \sum_{k=1}^{c} \sum_{i \in I_k} \max\left(0, 1 - \frac{\rho_w(x_i, k)}{\rho_k}\right) \right], \tag{16}$$

where, as in the binary classification, $\rho_k$s are chosen via cross-validation. While $\rho_k$s can be chosen freely, the analysis of lemma F.1 suggests concentrating the search around $r = \left[ \dfrac{m_k^{\frac{1}{3}} r_{k,2}^{\frac{2}{3}}}{\bar{r}^{\frac{2}{3}}} \right]_{k \in [c]}$.

The above can be generalized to other multi-class surrogate loss functions. In particular, when using the cross-entropy loss function applied to the outputs of a neural network, the (multinomial) logistic loss, our algorithm has the following form:

$$\min_{w \in \mathbb{R}^d} \lambda \|w\|^2 + \frac{1}{m} \sum_{k=1}^{c} \sum_{i \in I_k} \log\left[ 1 + \sum_{k' \neq k} e^{\frac{h(x_i, k') - h(x_i, k)}{\rho_k}} \right]. \tag{17}$$

where $\rho_k$s are chosen via cross-validation. When the number of classes $c$ is large, we can restrict our search by considering the same $\rho_k$ for classes with small representation, and distinct $\rho_k$s for the top classes. Similar algorithms can be devised for other $\|\cdot\|_p$ upper bounds on $w$, with $p \in [1, \infty)$. We can also derive a group-norm based generalization guarantee and corresponding algorithm.

### F.7. Proof of Theorem F.3 and Theorem F.4

**Theorem F.3.** *Consider* $\mathcal{H}_1 = \left\{ (x, y) \mapsto w \cdot \Phi(x, y) : w \in \mathbb{R}^d, \|w\|_1 \leq \Lambda_1 \right\}$. *Let* $r_{k,\infty} = \sup_{i \in I_k, y \in \mathcal{Y}} \|\Phi(x_i, y)\|_\infty$, *for any* $k \in [c]$. *Then, the following bound holds for all* $h \in \mathcal{H}$:

$$\widehat{\mathfrak{R}}_S^\rho(\mathcal{H}_1) \leq \frac{\Lambda_1 \sqrt{2c}}{m} \sqrt{\sum_{k=1}^{c} \frac{m_k r_{k,\infty}^2}{\rho_k^2} \log(2d)}.$$

*Proof.* The proof follows through a series of inequalities:

$$\widehat{\mathfrak{R}}_S^\rho(\mathcal{H}_1)$$
$$= \frac{1}{m} \mathbb{E}_\epsilon \left[ \sup_{\|w\|_1 \leq \Lambda_1} w \cdot \left( \sum_{k=1}^{c} \sum_{i \in I_k} \sum_{y \in \mathcal{Y}} \epsilon_{iy} \frac{\Phi(x_i, y)}{\rho_k} \right) \right]$$
$$\leq \frac{\Lambda_1}{m} \mathbb{E}_\epsilon \left[ \left\| \sum_{k=1}^{c} \sum_{i \in I_k} \sum_{y \in \mathcal{Y}} \epsilon_{iy} \frac{\Phi(x_i, y)}{\rho_k} \right\|_\infty \right] = \frac{\Lambda_1}{m} \mathbb{E}_\epsilon \left[ \max_{j \in [d], s \in \{-1, +1\}} s \sum_{k=1}^{c} \sum_{i \in I_k} \sum_{y \in \mathcal{Y}} \epsilon_{iy} \frac{\Phi_j(x_i, y)}{\rho_k} \right]$$
$$\leq \frac{\Lambda_1}{m} \left[ 2c \left( \sum_{k=1}^{c} \frac{m_k r_{k,\infty}^2}{\rho_k^2} \right) \log(2d) \right]^{\frac{1}{2}} = \frac{\Lambda_1 \sqrt{2c}}{m} \sqrt{\sum_{k=1}^{c} \frac{m_k r_{k,\infty}^2}{\rho_k^2} \log(2d)}.$$

The first inequality makes use of Hölder's inequality and the bound on $\|w\|_1$, and the second one follows from the maximal inequality and the fact that a Rademacher variable is 1-sub-Gaussian, and $\sup_{i \in I_k, y \in \mathcal{Y}} \|\Phi(x_i, y)\|_\infty = r_{k,\infty}$. $\qquad \square$

**Theorem F.4.** *Consider* $\mathcal{H}_2 = \left\{ (x, y) \mapsto w \cdot \Phi(x, y) : w \in \mathbb{R}^d, \|w\|_2 \leq \Lambda_2 \right\}$. *Let* $r_{k,2} = \sup_{i \in I_k, y \in \mathcal{Y}} \|\Phi(x_i, y)\|_2$, *for any* $k \in [c]$. *Then, the following bound holds for all* $h \in \mathcal{H}$:

$$\widehat{\mathfrak{R}}_S^\rho(\mathcal{H}_2) \leq \frac{\Lambda_2 \sqrt{c}}{m} \sqrt{\sum_{k=1}^{c} \frac{m_k r_{k,2}^2}{\rho_k^2}}.$$

*Proof.* The proof follows through a series of inequalities:

$$\widehat{\mathfrak{R}}_S^\rho(\mathcal{H}_2)$$

$$= \frac{1}{m} \mathbb{E}_\epsilon \left[ \sup_{\|w\|_2 \leq \Lambda_2} w \cdot \left( \sum_{k=1}^c \sum_{i \in I_k} \sum_{y \in \mathcal{Y}} \epsilon_{iy} \frac{\Phi(x_i, y)}{\rho_k} \right) \right]$$

$$\leq \frac{\Lambda_2}{m} \mathbb{E}_\epsilon \left[ \left\| \sum_{k=1}^c \sum_{i \in I_k} \sum_{y \in \mathcal{Y}} \epsilon_{iy} \frac{\Phi(x_i, y)}{\rho_k} \right\|_2 \right] \leq \frac{\Lambda_2}{m} \left[ \mathbb{E}_\epsilon \left[ \left\| \sum_{k=1}^c \sum_{i \in I_k} \sum_{y \in \mathcal{Y}} \epsilon_{iy} \frac{\Phi(x_i, y)}{\rho_k} \right\|_2^2 \right] \right]^{\frac{1}{2}}$$

$$\leq \frac{\Lambda_2}{m} \left[ \sum_{k=1}^c \frac{1}{\rho_k^2} \sum_{i \in I_k} \sum_{y \in \mathcal{Y}} \|\Phi(x_i, y)\|_2^2 \right]^{\frac{1}{2}} \leq \frac{\Lambda_2}{m} \sqrt{c \sum_{k=1}^c \frac{m_k r_{k,2}^2}{\rho_k^2}} = \frac{\Lambda_2 \sqrt{c}}{m} \sqrt{\sum_{k=1}^c \frac{m_k r_{k,2}^2}{\rho_k^2}}.$$

The first inequality makes use of the Cauchy-Schwarz inequality and the bound on $\|w\|_2$, the second follows by Jensen's inequality, the third by $\mathbb{E}[\epsilon_{iy}\epsilon_{jy'}] = \mathbb{E}[\epsilon_{iy}]\mathbb{E}[\epsilon_{jy'}] = 0$ for $i \neq j$ and $y \neq y'$, and the fourth one by $\sup_{i \in I_k, y \in \mathcal{Y}} \|\Phi(x_i, y)\|_2 = r_{k,2}$. $\qquad \square$

