# OpenReview forum: "Balancing the Scales: A Theoretical and Algorithmic Framework for Learning from Imbalanced Data"
_ICML.cc/2025/Conference — ICML 2025 poster_

### Official Review · Reviewer_Rew5 · 2025-02-24

**Overall Recommendation:** 4

**Summary:**

This paper presents a theoretical and algorithmic framework for addressing the class imbalance problem in machine learning, particularly in multi-class settings with long-tailed distributions. The authors introduce a novel class-imbalanced margin loss function for both binary and multi-class classification, proving its strong H-consistency and deriving learning guarantees based on empirical loss and class-sensitive Rademacher complexity. They propose a new algorithm, IMMAX (Imbalanced Margin Maximization), which incorporates confidence margins and is applicable to various hypothesis sets. The paper also provides extensive empirical results demonstrating the effectiveness of IMMAX compared to existing baselines on benchmark datasets like CIFAR-10, CIFAR-100, and Tiny ImageNet.

**Claims And Evidence:**

Yes, the claims are supported by a clear presentation and rigorous theoretical explanation.

**Essential References Not Discussed:**

The key contribution includes presenting a comprehensive theoretical analysis of generalization for classification loss in the context of imbalanced classes. The authors state that "only (Cao et al., 2019) provides an analysis of generalization guarantees, which is limited to
the balanced loss, the uniform average of misclassification errors across classes. Their analysis also applies only to binary classification under the separable case and does not address the target misclassification loss.". However, [1] has extended Cao's analysis to multiclass scenarios. Moreover, recent advances [2] also provide a fine-grained and tighter generalization guarantee for re-weighting and loss adjustment.  I strongly suggest the authors provide some more essential discussion.

[1] Balanced Meta-Softmax for Long-Tailed Visual Recognition. NeurIPS 2020.

[2] A Unified Generalization Analysis of Re-Weighting and Logit-Adjustment for Imbalanced Learning. NeurIPS 2023.

**Experimental Designs Or Analyses:**

The authors follow a rigorous experimental setup, using standard data augmentations and training procedures. The results are averaged over multiple runs, and standard deviations are reported, ensuring the reliability of the findings. The experiments cover a range of imbalance ratios, demonstrating the effectiveness of IMMAX in different scenarios.

**Methods And Evaluation Criteria:**

Yes, the proposed method make sense, and the evaluation criteria follows the common practice.

**Other Comments Or Suggestions:**

I hope the authors can address the aforementioned weaknesses and do not have more comments.

**Other Strengths And Weaknesses:**

Strengths:
- **Strong Theoretical Foundation**: The authors introduce a novel class-imbalanced margin loss function and provide detailed proofs of its H-consistency. The paper not only proposes a new loss function but also derives generalization bounds based on empirical loss and class-sensitive Rademacher complexity. These theoretical guarantees are crucial for understanding why the proposed method works and under what conditions it can be expected to perform well.
- **Novelty and Innovation**: The introduction of class-sensitive Rademacher complexity is a novel and innovative contribution. This concept allows the authors to derive generalization bounds that explicitly account for class imbalance, which is a key challenge in imbalanced learning. By incorporating confidence margins into the loss function, the authors address the limitations of existing methods, such as their tendency to overfit minority classes or discard valuable information from majority classes.

Weaknesses:
- **Insufficient citation of relevant literature**： The paper asserts that only one study has analyzed generalization guarantees. However, this overlooks several other significant works [1,2]. In particular, [2] offers insights that align closely with your research and analyzes existing reweighting methods by deriving tighter bounds. It might be beneficial to reference these works and compare their approaches with your findings to enrich the discussion.
- **Lack of empirical analysis on the hyperparameter $\rho_k$**： The IMMAX loss function introduces a hyperparameter for each class, which could pose significant tuning challenges, especially in large-scale datasets. The presence of numerous hyperparameters may also lead to unstable outcomes. A thorough empirical analysis is crucial to understand the impact and manageability of these hyperparameters. Conducting tests on a widely recognized dataset like ImageNet, which contains 1000 labels, would be highly beneficial to assess the scalability and robustness of the proposed method.

[1] Balanced Meta-Softmax for Long-Tailed Visual Recognition. NeurIPS 2020.

[2] A Unified Generalization Analysis of Re-Weighting and Logit-Adjustment for Imbalanced Learning. NeurIPS 2023.

**Questions For Authors:**

Does the proposed IMMAX loss function work like a class-wise temperature scaling technique based on the CE loss? Can I interpret it this way?
For other questions, please refer to Weakness.

**Relation To Broader Scientific Literature:**

The paper is well-situated within the broader literature on class imbalance in machine learning. The authors discuss various existing approaches, including data modification methods, cost-sensitive techniques, and logistic loss modifications. They highlight the limitations of these methods, particularly their lack of theoretical foundations and Bayes inconsistency. The proposed framework addresses these limitations by providing a principled approach to learning from imbalanced data supported by strong theoretical guarantees.

**Theoretical Claims:**

Yes, I have checked the proof, and the theoretical claims are sound and well-justified.

---

> ### Author Rebuttal · Authors · 2025-03-30
>
> Thank you for your encouraging review. We will take your suggestions into account when preparing the final version. Please find responses to your specific questions below.
>
> **1. Essential References Not Discussed: The key contribution ... However, [1] has extended Cao's analysis to multiclass scenarios. Moreover, recent advances [2] also provide a fine-grained and tighter generalization guarantee for re-weighting and loss adjustment. I strongly suggest the authors provide some more essential discussion.**
>
> **Weaknesses 1. Insufficient citation of relevant literature: The paper asserts that only one study has analyzed generalization guarantees. However, this overlooks several other significant works [1,2]. In particular, [2] offers insights that align closely with your research and analyzes existing reweighting methods by deriving tighter bounds. It might be beneficial to reference these works and compare their approaches with your findings to enrich the discussion.**
>
> **Response:** Thank you very much for bringing these references to our attention. They are indeed important contributions to the analysis of generalization guarantees in imbalanced learning. Briefly, these works focus on generalization with respect to the balanced loss, whereas our work addresses generalization guarantees with respect to the standard zero-one misclassification loss. We will include a more detailed discussion and comparison in the final version.
>
> **2. Weaknesses 2. Lack of empirical analysis on the hyperparameter: The IMMAX loss function introduces a hyperparameter for each class, which could pose significant tuning challenges, especially in large-scale datasets. The presence of numerous hyperparameters may also lead to unstable outcomes. A thorough empirical analysis is crucial to understand the impact and manageability of these hyperparameters. Conducting tests on a widely recognized dataset like ImageNet, which contains 1000 labels, would be highly beneficial to assess the scalability and robustness of the proposed method.**
>
> **Response:** The number of hyperparameters is indeed an important consideration. As discussed at the end of Section 5, when the number of classes is very large, the search space can be significantly reduced by assigning identical $\rho_k$ values to underrepresented classes while reserving distinct values for the most frequently occurring ones.
>
> Moreover, while $\rho_k$ values can be freely searched, the search can be guided by the vector $[\rho_k / \sum_{k = 1}^c \rho_k]_k$ near $\mathbf{r}$, which corresponds to the theoretically optimal values in the separable case. This approach was also adopted in our experiments.
>
> We have not observed instability issues in our experiments. However, as suggested by the reviewer, we will include additional experimental results in the final version to further study this point empirically. ImageNet would indeed be an interesting dataset to try.
>
> **3. Questions: Does the proposed IMMAX loss function work like a class-wise temperature scaling technique based on the CE loss? Can I interpret it this way?**
>
> **Response:** Yes, one could interpret it as a class-based temperature scaling derived from the logistic loss. However, our choice is grounded in a theoretical argument that justifies its ability to establish distinct confidence margins across classes, as elaborated in our analysis. In fact, our theoretical framework could provide an insightful interpretation and justification for the familiar temperature parameters.

---

> > ### Comment · Reviewer_Rew5 · 2025-04-03
> >
> > The response has addressed my concern, and I will increase my score to 4.

---

### Official Review · Reviewer_aY6w · 2025-03-10

**Overall Recommendation:** 4

**Summary:**

The first main result is a consistency bound for a class-imbalanced margin loss when the hypothesis set is complete. The second result is a margin-based generalization bound for imbalanced binary classification in terms of Rademacher complexity. The last result is a bound for the Rademacher complexity when the hypothesis set is a class of linear hypotheses with bounded weight vectors. The analysis is also extended to multi-class classification.

**Claims And Evidence:**

The claims are supported by clear and convincing theoretical results. The results are nice. It is enjoyable to read the paper.
The first main result, Theorem 3.3, is given when the hypothesis set is complete. Here the completeness is essential for the bound. The last result, Theorem 4.1, is stated only when the hypothesis set is a class of linear hypotheses with bounded weight vectors. This is rather special, though some hypothesis sets generated by deep neural networks with bounded Frobenius norms or spectral norms are covered. It would be better if more general hypothesis sets generated by deep neural networks could be considered. Moreover, the authors might consider consistency bounds for a class-imbalanced margin loss when the hypothesis set is not complete, especially those corresponding to Theorem 4.1, with uniformly bounded hypotheses.

**Essential References Not Discussed:**

Consistent bounds for imbalanced binary classification have been well studied, much earlier than the first reference. One can find such results in the literature of Zhang (Ann. Stat. 2004), Bartlett-Jordan-McAuliffe (JASA 2006), Chen-Wu-Ying-Zhou (JMLR 2004).

**Experimental Designs Or Analyses:**

The experimental designs seem reasonable.

**Methods And Evaluation Criteria:**

The methods used for the theoretical study in the paper are based on functional analysis for the related generalization error and 0-1 error and Rademacher analysis. These are appropriate and convincing. If more approximation theory or deep neural network analysis could be used, they would lead to further research activities in dealing with nonlinear or bounded hypotheses.

**Other Comments Or Suggestions:**

It would be better if more general hypothesis sets generated by deep neural networks could be considered. Moreover, the authors might consider consistency bounds for a class-imbalanced margin loss when the hypothesis set is not complete, especially those corresponding to Theorem 4.1, with uniformly bounded hypotheses.

**Other Strengths And Weaknesses:**

none.

**Questions For Authors:**

none.

**Relation To Broader Scientific Literature:**

The key contributions of the paper are about imbalanced data which appear in many practical applications. They can be useful for theory of fair machine learning, a timely and important topic.

**Theoretical Claims:**

Yes, I check the correctness of the proofs. The proofs are correct but are pretty easy to give, which is suitable for a conference paper.

---

> ### Author Rebuttal · Authors · 2025-03-30
>
> Thank you for your appreciation of our work. We will take your suggestions into account when preparing the final version. Below please find responses to specific questions.
>
> **1. Methods And Evaluation Criteria: The methods used for the theoretical study in the paper are based on functional analysis for the related generalization error and 0-1 error and Rademacher analysis. These are appropriate and convincing. If more approximation theory or deep neural network analysis could be used, they would lead to further research activities in dealing with nonlinear or bounded hypotheses.**
>
> **Response:** To clarify, the margin-based generalization bounds we presented also apply to families of neural networks. The reviewer is correct that a deeper analysis from the perspective of approximation theory could further complement our results. However, this is a broader question that extends to other types of learning guarantees as well.
>
> **2. Essential References Not Discussed: Consistent bounds for imbalanced binary classification have been well studied, much earlier than the first reference. One can find such results in the literature of Zhang (Ann. Stat. 2004), Bartlett-Jordan-McAuliffe (JASA 2006), Chen-Wu-Ying-Zhou (JMLR 2004).**
>
> **Response:** Thank you for your suggestion. We will add these references. However, we note that these studies focus on Bayes consistency in standard binary classification, rather than specifically addressing the imbalanced setting.
>
> **3. Other Comments Or Suggestions: It would be better if more general hypothesis sets generated by deep neural networks could be considered. Moreover, the authors might consider consistency bounds for a class-imbalanced margin loss when the hypothesis set is not complete, especially those corresponding to Theorem 4.1, with uniformly bounded hypotheses.**
>
> **Response:** Thank you for the suggestions. Our H-consistency bounds in Theorem 3.3 can indeed be extended to the uniformly bounded hypothesis sets considered in Theorem 4.1. In this case, the bounds would depend on the complexity of the hypothesis class, similar to the H-consistency bounds presented in [1]. We will include this extension in the final version.
>
> We would be happy to discuss extensions to other neural network families the reviewer might suggest.
>
> [1] Awasthi et al. H-Consistency Bounds for Surrogate Loss Minimizers. ICML 2022.

---

### Official Review · Reviewer_H6YG · 2025-03-12

**Overall Recommendation:** 4

**Summary:**

This paper introduces a novel theoretical framework for analyzing generalization in imbalanced classification. It proposes a new class-imbalanced margin loss function for both binary and multi-class settings, proves its strong $\mathcal{H}$-consistency, and derives corresponding learning guarantees based on empirical loss and a new notion of class-sensitive Rademacher complexity. It then devises novel and general learning algorithms, IMMAX (Imbalanced Margin Maximization), which incorporate confidence margins and are applicable to various hypothesis sets. Experiments demonstrate the effectiveness of the proposed method.

## update after rebuttal
The authors' rebuttal addresses concerns. I would like to keep my rating and support the acceptance of the paper.

**Claims And Evidence:**

The claims are supported by theoretical analysis and experimental verification.

**Essential References Not Discussed:**

The paper cites key literature in the field of imbalanced learning, including data modification techniques, cost-sensitive methods, and logistic loss modifications. However, it could benefit from discussing more recent advances in deep learning-based approaches for imbalanced data, such as those involving neural network architectures specifically designed to handle class imbalance.

**Experimental Designs Or Analyses:**

The experimental designs are sound and demonstrate the effectiveness of the proposed methods.

**Methods And Evaluation Criteria:**

The proposed methods and evaluation criteria make sense for the studied problem. The IMMAX algorithm incorporates confidence margins and is applicable to various hypothesis sets, and the evaluation metrics are appropriate for assessing the performance of the proposed method.

**Other Comments Or Suggestions:**

It is suggested to add the top and bottom lines in Tables 1, 2, and 3 to make them more intuitive.

**Other Strengths And Weaknesses:**

### Strengths

- The problem studied in this paper is interesting.

- This paper is well written and in good sharp, which is easy to follow.

- The experimental results are somehow promising.

- The theoretical work and empirical studies of this paper are sufficient, which improves the value of the paper.



### Weaknesses

- The performance of IMMAX in large-scale data sets is not clear, which is the key to its application in practical scenarios.

**Questions For Authors:**

How does the IMMAX algorithm scale with increasing dataset size and dimensionality? Are there any computational limitations that might affect its practicality for very large datasets?

**Relation To Broader Scientific Literature:**

The paper's contributions are well-related to the broader scientific literature on class imbalance in machine learning.

**Theoretical Claims:**

The theoretical claims are correctly established. The authors provide detailed proofs and discussions of the theoretical results, demonstrating a solid understanding of the underlying principles.

---

> ### Author Rebuttal · Authors · 2025-03-30
>
> Thank you for your appreciation of our work. We will take your suggestions into account when preparing the final version. Below please find responses to specific questions.
>
> **1. Essential References Not Discussed: The paper cites key literature in the field of imbalanced learning, including data modification techniques, cost-sensitive methods, and logistic loss modifications. However, it could benefit from discussing more recent advances in deep learning-based approaches for imbalanced data, such as those involving neural network architectures specifically designed to handle class imbalance.**
>
> **Response:** We aimed to provide a comprehensive overview of related work given the space constraints. We would be happy to expand our discussion to include architecture-based solutions, some of which are already covered in the survey paper we referenced. If the reviewer has specific publications in mind, we would gladly incorporate and discuss them in more detail.
>
> **2. Weaknesses: The performance of IMMAX in large-scale datasets is not clear, which is the key to its application in practical scenarios.**
>
> **Questions: How does the IMMAX algorithm scale with increasing dataset size and dimensionality? Are there any computational limitations that might affect its practicality for very large datasets?**
>
> **Response:** The dependency of our solution on sample size and dimensionality is similar to that of standard neural networks trained with cross-entropy loss (that is the logistic loss when softmax is applied to logits). Thus, our approach remains practical when using optimizers such as SGD, Adam, or AdaGrad. Our solution does depend on the number of classes, but this dependency is inherent to standard multi-class neural network solutions as well.
>
> **3. Other Comments Or Suggestions: It is suggested to add the top and bottom lines in Tables 1, 2, and 3 to make them more intuitive.**
>
> **Response:** Thank you for the suggestion. We’ll add the lines to the tables in the final version.

---

### Official Review · Reviewer_zC71 · 2025-03-13

**Overall Recommendation:** 4

**Summary:**

The paper addresses the challenge of class imbalance in machine learning, particularly in multi-class problems with long-tailed distributions. The authors propose a novel theoretical framework for analyzing generalization in imbalanced classification, introducing a class-imbalanced margin loss function for both binary and multi-class settings. They prove the strong H-consistency of this loss function and derive learning guarantees based on empirical loss and a new notion of class-sensitive Rademacher complexity. Leveraging these theoretical results, the authors devise the IMMAX algorithm, which incorporates confidence margins and is applicable to various hypothesis sets. The paper also presents extensive empirical results demonstrating the effectiveness of IMMAX compared to existing baselines.

**Claims And Evidence:**

The claims made in the paper are generally supported by clear and convincing evidence. The authors provide rigorous theoretical proofs for the H-consistency of their proposed class-imbalanced margin loss function and derive generalization bounds based on class-sensitive Rademacher complexity.

**Essential References Not Discussed:**

No.

**Experimental Designs Or Analyses:**

The experimental design is sound and valid. However, I have summarized some questions in the weaknesses and questions sections.

**Methods And Evaluation Criteria:**

The proposed methods and evaluation criteria are appropriate for the problem of class imbalance in machine learning. The authors use standard benchmark datasets and compare their algorithm against several well-known baselines, including cross-entropy loss, Re-Weighting, Balanced Softmax, and LDAM loss.

**Other Comments Or Suggestions:**

In the paper, the meaning of  $\rho$  appears to be ambiguous, as it is redefined in line 366.

**Other Strengths And Weaknesses:**

Strengths:
- The paper provides a rigorous theoretical framework for addressing class imbalance, which is a significant contribution to the field.
- The proposed IMMAX algorithm is general and can be applied to various hypothesis sets, making it a versatile solution for imbalanced classification problems.
- The empirical results are convincing and demonstrate the effectiveness of the proposed method across multiple datasets and imbalance ratios.

Weaknesses:
- The experimental results presented in the paper (including the results of the comparison methods) are significantly better than those reported in previous papers. Could the authors provide further explanation for this discrepancy? For example, are there differences in the experimental setup, data preprocessing, or evaluation metrics that could account for the improved performance?
- The selection of $\rho$ appears to be based on the validation set. However, the paper does not explain how the validation set was constructed. Given that one of the challenges in imbalanced learning is the scarcity of samples in the minority class, it may be difficult to obtain a sufficient number of samples for a reliable validation set. Could the authors clarify how the validation set was constructed and how they ensured its representativeness? Additionally, $\rho$ do not seem to provide a directly applicable, general parameter prior, which could limit the practical application of the method. Could the authors discuss potential strategies for selecting these parameters in real-world scenarios where validation data may be limited?

**Questions For Authors:**

1. The results in Table 1 appear to differ significantly from those reported in previous papers. For example, in the CIFAR-10 dataset with a ratio of 100, the accuracy of Balanced Softmax (BS) is typically around 80, whereas in this paper, it reaches 95. This discrepancy is quite unusual. Similarly, the experimental results on other datasets and settings also seem to be generally higher. Could the authors provide an explanation for these differences?
2. The IMMAX loss function differs significantly in form from Softmax. In contrast, methods like Balanced Softmax (BS), Logit Adjusted (LA), and LDAM loss revert to the standard softmax cross-entropy loss when the training dataset is balanced. However, IMMAX does not exhibit this behavior. This seems somewhat unreasonable. If the training dataset were balanced, would IMMAX perform better than the standard softmax cross-entropy loss?
3. IMMAX seems more akin to a contrastive loss. Could it be applied in a supervised contrastive learning scenario? If so, how would it compare to existing supervised contrastive learning methods?

**Relation To Broader Scientific Literature:**

The key contributions of the paper are well-aligned with the broader scientific literature on class imbalance in machine learning. The authors build on existing work on data resampling, cost-sensitive techniques, and logistic loss modifications, providing a more principled theoretical foundation for these methods.

**Theoretical Claims:**

I am not particularly familiar with the relevant theories, so I am unable to assess the correctness of the theoretical proofs.

---

> ### Author Rebuttal · Authors · 2025-03-30
>
> Thank you for your appreciation of our work. We will take your suggestions into account when preparing the final version. Below please find responses to specific questions.
>
> **1. Weaknesses 1: The experimental results presented in the paper (including the results of the comparison methods) are significantly better than those reported in previous papers. Could the authors provide further explanation for this discrepancy? For example, are there differences in the experimental setup, data preprocessing, or evaluation metrics that could account for the improved performance?**
>
> **Question 1: The results in Table 1 appear to differ significantly from those reported in previous papers ... Could the authors provide an explanation for these differences?**
>
> **Response:** Our work focuses on the standard and unmodified zero-one misclassification loss, which remains the primary objective in many machine learning applications, as discussed in the introduction. Accordingly, we report standard accuracy based on this loss function. In contrast, some previous studies report "balanced accuracy," which averages misclassification errors uniformly across classes (i.e., the balanced loss). This difference in evaluation metrics explains the higher values reported in our results. The balanced accuracy of Balanced Softmax (BS) on CIFAR-10 with a ratio of 100 in our experimental setting is also around 80%. We will provide further elaboration in the final version.
>
> Regarding the experimental setup and data preprocessing, we strictly followed the procedure of Cao et al. (2019), ensuring consistency in all these aspects.
>
> **2. Weaknesses 2: The selection of $\rho$ appears to be based on the validation set ... Could the authors discuss potential strategies for selecting these parameters in real-world scenarios where validation data may be limited?**
>
> **Response:** We tune the hyperparameters using a validation set held out separately from the training set. Additional details on cross-validation are provided in Appendix B, and further elaboration will be included in the final version. Empirically, performance is not sensitive to variations in the neighborhood of the theoretically optimal values of the $\rho_k$s indicated below.
>
> As discussed at the end of Section 5 (Lines 302-309, second column), while the $\rho_k$ values can be freely searched over a range of values in our general algorithm, the search can be guided by the vector $[\rho_k / \sum_{k = 1}^c \rho_k]_k$ near $\mathbf{r}$, which corresponds to the theoretically optimal values of $\rho_k$ in the separable case. We adopted this approach in our experiments. Moreover, in scenarios with a large number of classes, the search space can be significantly reduced by assigning identical $\rho_k$ values to underrepresented classes, while reserving distinct $\rho_k$ values for the most frequently occurring classes. This strategy enhances practicality when validation data is limited, with only a minor impact on results.
>
> **3. Question 2: The IMMAX loss function differs significantly in form from Softmax ... If the training dataset were balanced, would IMMAX perform better than the standard softmax cross-entropy loss?**
>
> **Response:** When the training dataset is balanced, the theoretically optimal values of $\rho_k$ are identical across all classes. In this case, IMMAX becomes equivalent to the standard softmax cross-entropy loss with an appropriate regularization parameter. Therefore, IMMAX would perform similarly to the standard softmax cross-entropy loss in the balanced setting.
>
> **4. Question 3: IMMAX seems more akin to a contrastive loss. Could it be applied in a supervised contrastive learning scenario? If so, how would it compare to existing supervised contrastive learning methods?**
>
> **Response:** The form of our loss function has some similarity with supervised contrastive losses (e.g., [1]), where a scalar temperature parameter is used in the inner product argument of the exponential. However, in our case, distinct parameters are introduced to allow different confidence margins across classes, serving a different purpose than in contrastive learning. Nevertheless, our margin analysis could provide a useful tool for analyzing contrastive learning. We will acknowledge this connection, include a brief discussion, and thank the reviewer for the suggestion.
>
> [1] Khosla et al. Supervised Contrastive Learning. NeurIPS 2020.
>
> **5. Other Comments Or Suggestions: In the paper, the meaning of $\rho$ appears to be ambiguous, as it is redefined in line 366.**
>
> **Response:** Thank you for pointing this out. We will change the notation here to avoid any overlap with our confidence margin definition of $\rho$ throughout the paper.

---

### Decision · Program_Chairs · 2025-05-01

**Decision:**

Accept (poster)

**Comment:**

This paper introduces a new theoretical framework for addressing class imbalance in both binary and multi-class classification tasks, filling a critical gap in existing methods that lack rigorous foundations. It proposes a novel class-imbalanced margin loss function with strong H-consistency guarantees and derives principled generalization bounds via a new notion of class-sensitive Rademacher complexity. Building on this foundation, the authors present IMMAX， a family of margin-based learning algorithms that are broadly applicable and outperform many established baselines. Unlike traditional cost-sensitive or resampling methods, this approach is Bayes-consistent and analytically grounded. Extensive empirical results further demonstrate the effectiveness and robustness of the proposed algorithms.

During the rebuttal phase, the authors addressed the raised concerns, resulting in improved reviewer scores from (4, 4, 4, 3) to (4, 4, 4, 4). Therefore, I recommend accepting this paper. Notably, the manuscript would benefit from further revision, especially in elaborating on recent theoretical advances and the role of $\rho$.